# Drug screening on digital microfluidics for cancer precision medicine

Jiao Zhai[1,2,13], Yingying Liu[1,3,13], Weiqing Ji[4,13], Xinru Huang[5], Ping Wang[6], Yunyi Li[1], Haoran Li [1,3], Ada Hang-Heng Wong [7], Xiong Zhou[1,8], Ping Chen[9], Lianhong Wang[8], Ning Yang[1,10], Chi Chen[5], Haitian Chen [5], Pui-In Mak[1,3], Chu-Xia Deng [9], Rui Martins[1,3,11], Mengsu Yang [2], Tsung-Yi Ho [12], Shuhong Yi [5,14] ✉, Hailong Yao [4,14] ✉ & Yanwei Jia [1,3,7,14] ✉

Drug screening based on in-vitro primary tumor cell culture has demonstrated potential in personalized cancer diagnosis. However, the limited number of tumor cells, especially from patients with early stage cancer, has hindered the widespread application of this technique. Hence, we developed a digital microfluidic system for drug screening using primary tumor cells and established a working protocol for precision medicine. Smart control logic was developed to increase the throughput of the system and decrease its footprint to parallely screen three drugs on a 4 × 4 cm² chip in a device measuring 23 × 16 × 3.5 cm³. We validated this method in an MDA-MB-231 breast cancer xenograft mouse model and liver cancer specimens from patients, demonstrating tumor suppression in mice/patients treated with drugs that were screened to be effective on individual primary tumor cells. Mice treated with drugs screened on-chip as ineffective exhibited similar results to those in the control groups. The effective drug identified through on-chip screening demonstrated consistency with the absence of mutations in their related genes determined via exome sequencing of individual tumors, further validating this protocol. Therefore, this technique and system may promote advances in precision medicine for cancer treatment and, eventually, for any disease.

Precision medicine in oncology is needed to tailor therapeutic strategies to individual patients with cancer and ensure the best prognosis[1–3]. To date, most precision medicines are based on genetic abnormalities of each patient. Some drugs can be prescribed to patients with certain genetic mutations for an optimal response, whereas other patients with specific mutations are not prescribed these drugs due to predicted reduced responsiveness or a high risk of adverse effects[4–6]. However, clinical data have indicated that an increasing number of

[1]State Key Laboratory of Analog and Mixed-Signal VLSI, Institute of Microelectronics, University of Macau, Macau SAR, China. [2]Department of Biomedical Sciences, and Tung Biomedical Sciences Centre, City University of Hong Kong, Hong Kong SAR, China. [3]Faculty of Science and Technology, University of Macau, Macau SAR, China. [4]School of Computer and Communication Engineering, University of Science and Technology Beijing, Beijing, China. [5]Liver Transplantation Center, The Third Affiliated Hospital, Sun Yat-Sen University, Guangzhou, China. [6]Department of Hepatobiliary Surgery, The First Affiliated Hospital of Guangzhou Medical University, Guangzhou, China. [7]MoE Frontiers Science Center for Precision Oncology, University of Macau, Macau SAR, China. [8]College of electrical and information engineering, Hunan University, Changsha, China. [9]Cancer Center, Faculty of Health Sciences, University of Macau, Macau SAR, China. [10]Department of Electronic Information Engineering, Jiangsu University, Zhenjiang, China. [11]On leave from Instituto Superior Tecnico, Universidade de Lisboa, Lisboa, Portugal. [12]Department of Compute Science and Engineering, The Chinese University of Hong Kong, Hong Kong, China. [13]These authors contributed equally: Jiao Zhai, Yingying Liu, Weiqing Ji. [14]These authors jointly supervised this work: Shuhong Yi, Hailong Yao, Yanwei Jia. ✉e-mail: yishuhong@163.com; hailongyao@ustb.edu.cn; yanweijia@um.edu.mo

genes are involved in the cancer response to a certain drug, making the therapeutic effect based on genetic precision medicine unsatisfying[7]. Alternatively, drug screening of primary tumor cells from patient biopsies or tumor resection samples provides direct information on the drug susceptibility of the specific tumor[8,9]. However, biopsy samples contain only a limited number of cells (approximately $5 \times 10^4$ cells), making drug screening with traditional 96-well microplates difficult. Although multiple biopsies could provide enough tumor cells, it also raises the risk of cancer metastasis and the patient experiences more pain. The culture of primary tumor cells could generate enough cells for in-vitro drug screening. Nevertheless, the in-vitro culturing process may introduce unexpected mutations in daughter cells, leading to uncertain drug screening implications[10,11].

In recent years, microfluidics has emerged as a valuable tool in biomedical science for drug screening using primary tumor cells for precision medicine, with the ability to handle small samples. There are two main types of microfluidics: flow-based channel microfluidics and electric-based digital microfluidics. Drug screening has been performed on biopsies in channel microfluidics[8,9,12–15]. In these studies, human primary tumor samples from different organs, such as nasopharyngeal tumor[8], pancreatic cancer[9], and breast cancer[12] samples were used for drug sensitivity tests. For example, Wong et al. developed a polydimethylsiloxane (PDMS)-based droplet microfluidic platform to conduct drug screening of primary nasopharyngeal tumors from human patients. The results suggested diverse susceptibilities of primary nasopharyngeal tumors to different drugs[8]. Eduati et al. developed a two-phase Braille valve microfluidic platform for combinatorial drug screening of biopsies from clinical pancreatic cancers. They identified specific efficient drug combinations for each patient sample; however, no drug combinations had universal efficacy across all patients. This system provided higher throughput than that of other microfluidics systems by one to two orders of magnitude[9]. However, the long connecting tubes for sample loading in all the above channel microfluidic systems are associated with the waste of precious biopsy samples. Eduati et al. reported that ~100 live cells were required in a droplet, without mentioning the percentage of the input cells captured in droplets with drugs[9]. With channel microfluidics[16,17], the droplets are typically non-uniform for the first few minutes owing to unstable flow rate and pressures. Those droplets were normally discarded for analysis. To solve the problem of the swept volume introduced by syringes and tubing, Werner developed a microfluidic droplet generator system, which consisted of a series of peristaltic pumps controlled by an integrated pneumatic logic circuit, for reagents to be consumed directly from a well plate[18]. Nevertheless, the fabrication process was tedious and complicated. Furthermore, bulky supporting equipment and unfriendly operation protocols hinder their wide acceptance by doctors. The main drawback of wasting some biopsy samples for channel-based microfluidics makes it a hot potato, considering the limited precious biopsy samples. In contrast, the DMF system can manipulate a single droplet with precise control to use up all the biopsied cells. The convenience of DMF operation without the pumps or valves, as in channel microfluidics, also validates its advantages. In addition, the DMF technique facilitates the automated analysis of individual samples while occupying a much smaller footprint[19,20]. Cell culture on DMF chips has been explored by various groups for primary cell culture, single cell culture, or drug toxicity tests on commercial cancer cell lines or primary tumor cells[21–23]. Wheeler et al. developed an "upside-down" mode for primary cell culture in virtual microwells on a patterned top plate using DMF[19]. They successfully cultured aortic endothelial cells isolated from pig blood vessels, aortic valve endothelial cells, and aortic valve interstitial cells isolated from heart valves. In our previous study, we developed a DMF chip with 3D microstructures for single-cell drug screening and demonstrated that the $IC_{50}$ is comparable with those obtained off-chip on commercially available breast tumor cells and normal cells[23]. However, all these previous investigations used cancer cell lines or large amounts of primary cells isolated from organs as the model system for proof-of-principle validation. Primary cells isolated from organ model systems can undergo multiple sub-cultures as a cell line; thus, ideally any amount of cells could be achieved by culturing the cells to be used for drug screening on-chip. Direct drug screening on primary tumor cells ($<10^5$) without in-vitro subculture has never been tested using DMF. More importantly, whether in-vitro drug screening can provide valuable information to doctors about the different in-vivo reactions of individual patients to potentially effective drugs remains unclear. An integrated system friendly accessible to clinicians is in high demand for cancer precision medicine based on drug screening of primary tumor cells.

In this study, we established a DMF system for drug screening of primary tumor cells and tested it on MDA-MB-231 breast cancer xenograft mouse model and clinical liver cancer samples. Figure 1 shows a schematic representation of the system setup and the entire process, from sample preparation to data analysis of drug efficiency and precision medicine.

## Results

### Electrode-sharing logic on DMF chip
For precision medicine, the evaluation of cellular responses to various drugs requires fresh primary tumor cells, as they exhibit the most vital activity to demonstrate their responses to drugs. This requires immediate cell collection and on-site drug screening. A small-footprint drug screening system that is easy to operate is in high demand to handle such samples. To meet these requirements, we developed a portable digital microfluidic device that integrates all the controls into a handheld box with user-friendly control panels for the parallel screening of three drugs.

Figure S1a illustrates the device wrapped in a box measuring 23 (L) × 16 (W) × 3.5 (H) cm, which contained the electronic control and the DMF chip. The electric control circuit is schematically depicted in Fig. S1b, comprising a 5 V power supply, an electronic control PCB, a signal generator, a transformer, physical relays, and control button arrays. The DMF chip was on a glass substrate with electrode patterns the same as those in the control panel. The details of the chip design have been described in the SI, (Fig. S2).

To enhance the suitability of the DMF chip for high-throughput drug screening, we propose a smart electrode-sharing protocol for drug screening featuring a distinctive electrode connection structure and control algorithm.

Using the proposed protocol, 96 electrodes can be controlled on a 4 × 4 cm sized chip with 24 actuation signal channels. As the control panel was also minimized to the fewest relay number, the entire drug screening system can be miniaturized to a portable size, making it suitable for animal facilities or clinic spaces.

### Single drug screening of primary tumor samples from MDA-MB-231 breast cancer xenograft mouse model for precision medicine
Owing to its small footprint and high-throughput features, the DMF drug screening system is an ideal platform to screen drugs using a limited number of primary tumor cells from an individual patient, enabling precision medicine by identifying specific, effective drugs tailored for each patient. The number of primary tumor cells obtained with a biopsy needle was from $1 \times 10^4$ to $5 \times 10^4$ depending on the gauge of the biopsy needle (Fig. S3a, b). However, 96-well and 384-well plates require $1.5 \times 10^6$ cells or $4.0 \times 10^5$ cells, respectively, as previously reported[8] and based on experimental results (Fig. S3c). The required number of cells is markedly higher than that a biopsy sample can provide. The cell limit for drug screening on DMF chips can be as low as 100 cells per drug (Fig. S3d), rendering it a promising platform for drug screening of primary tumor samples. To validate the DMF drug

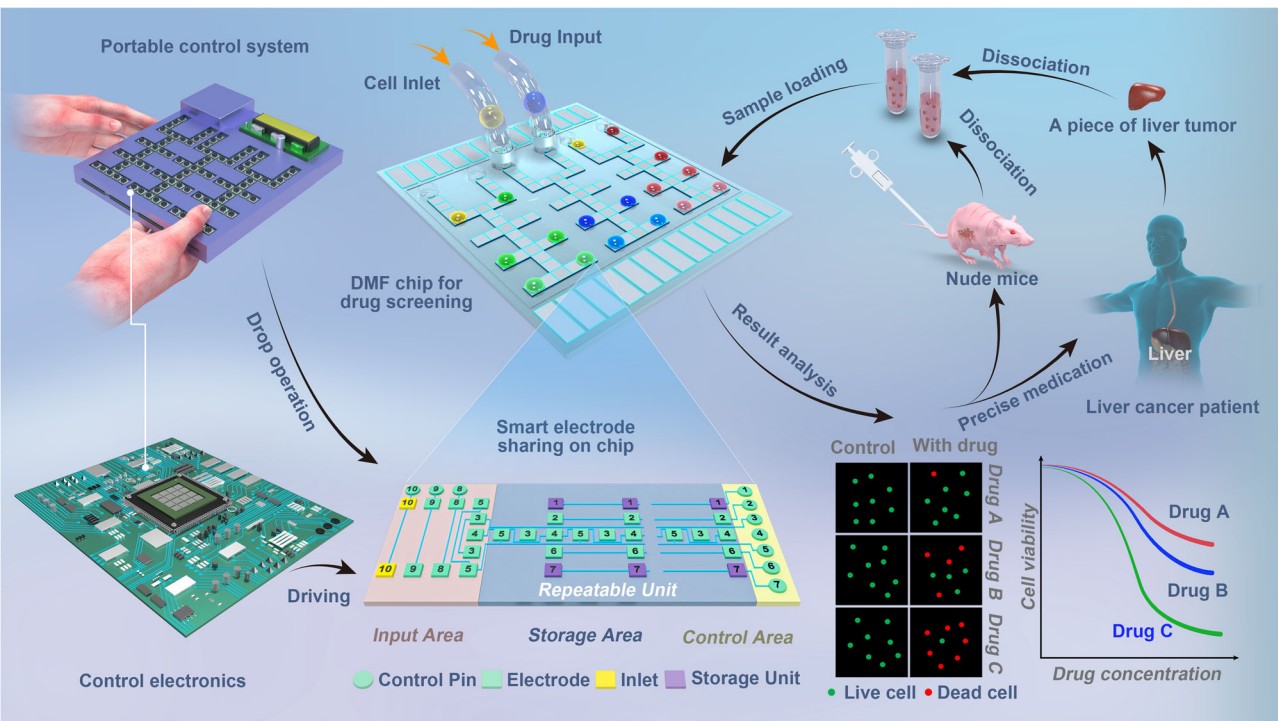

**Fig. 1 | Schematic of drug screening on digital microfluidics for cancer precision medicine.** Schematic of the digital microfluidic (DMF) system for drug screening of biopsy samples from MDA-MB-231 breast cancer xenograft mouse model and patients with liver cancer.

screening system for precision medicine, we conducted tests on an MDA-MB-231 breast cancer xenograft mouse model and treated them with common chemotherapeutic drugs.

The entire process from the MDA-MB-231 breast cancer xenograft mouse model to precision medicine is illustrated in the following figure (Fig. 3a, see below). Female nude mice were subcutaneously injected with a human breast cancer MDA-MB-231 cell suspension into the right flank to induce subcutaneous transplantation tumors. Once the tumors of most mice reached 0.1–0.3 cm³, the mice with appropriately sized tumors were weighed, and the tumors were analyzed for the initial parameters. Following anesthesia with avertin, the mice underwent a biopsy operation to remove some tumor samples using biopsy needles (Fig. 3b, see below). Each biopsy sample contained ~10,000 cells. The biopsy samples obtained were immediately dissociated for on-chip drug screening. The wound around the tumors of the mice was sealed with medical glue to prevent infection (Fig. 3c, see below). Three drugs were screened on-chip using each primary biopsy tumor sample from an individual mouse to determine which drug had the most toxic effect on a specific sample. The mice were subsequently sorted into three groups according to the in-vivo drug screening results. In the positive group, each mouse was injected with the most effective drug as determined on-chip for that specific mouse, whereas in the negative group, each mouse was injected with the least effective drug. The control group underwent the entire biopsy and injection processes, but the injection solution contained only PBS buffer. The detailed experimental protocol can be found in the SI.

In this experiment, cisplatin (Cis), Wzb117 (Wzb, the glucose transporter 1 inhibitor), and epirubicin hydrochloride (EP) were used as the drug models. Cis is a widely used anti-cancer chemotherapy drug[24,25], Wzb has been shown to block glucose transport, promote cell apoptosis, and suppress tumor growth in a xenograft mouse model[26–28], whereas EP possesses a wide spectrum of antitumor effects, showing efficacy for the treatment of metastatic breast cancer[29,30] and small cell lung cancer[31]. The primary tumor samples from 14 mice were

used for on-chip drug screening, with the same mice for the in-vivo therapy, either as the positive or negative group. Seven mice were treated in parallel with PBS buffer as a control.

The in vitro drug screening results are shown in Fig. 2a. The amount of biopsy samples from mouse #1, 8, and 9 were too little, which may be due to the half-filled biopsy needles or centrifugation process. The cells were enough for only two drug screenings. Each mouse responded differently to the same drug. For example, mouse #2 had the strongest response to Cis (cell viability was 47%, Cis 40 μM, Fig. 2b), whereas mouse #14 showed a reduced response to Cis (cell viability was 81%, Cis 40 μM, Fig. 2b). EP appeared ineffective (-) in mouse #9 but demonstrated effectiveness (cell viability was 55%, EP 40 μM, Fig. 2b) in mouse #11. Most mice showed no responses to Wzb. Representative images demonstrating the toxicity of the three drugs toward the primary tumor samples from mouse #6 are presented in Fig. 2c, with red fluorescence labeling indicating an increased number of dead cells at higher Cis or EP concentrations, whereas no significant change was observed in the presence of various concentrations of Wzb. There was no evident dependence of drug efficacy on mouse weight. However, larger tumors tended to show lower responses to a drug, possibly due to late-stage tumors harboring gene mutations that help tumor cells survive. By conducting drug screening on primary tumor cells and using the tumor immediate response to drugs rather than allowing for potential unknown gene mutations take place, it is possible to overcome the potential pitfalls of misdiagnosis that can hamper treatment.

To investigate whether the in-vitro primary tumor cell drug screening corresponded to the in-vivo drug responses, we studied the tumor growth by administering each mouse either an effective screened drug, an ineffective screened drug, or PBS as the control (Fig. 2b). The whole process (Fig. 3a) involved breast cancer MDA-MB-231 model establishment, biopsy sample collection (Fig. 3b), on-chip drug screening, and screening results guided treatment in vivo. Figure 3c showed the image of the mouse before and after obtaining a

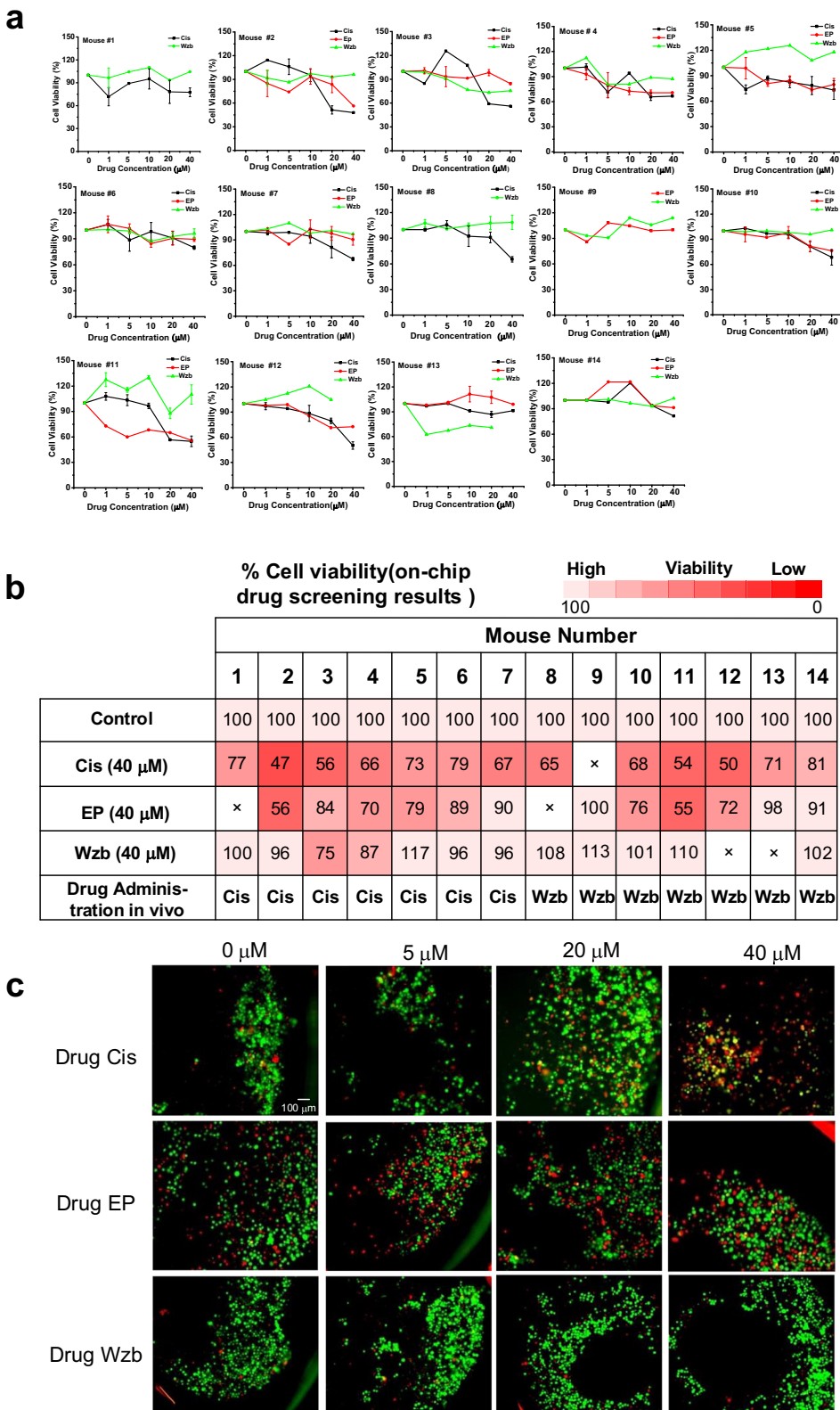

**Fig. 2 | Sing drug screening on chip. a** Cell viability results for the biopsy samples from 14 individual mice with different drugs, epirubicin hydrochloride (EP), cisplatin (Cis) and wzb117 (Wzb, the glucose transporter 1 inhibitor) treatment. mouse #1–5, *n* = 2; mouse #6, *n* = 3, mouse #7, 8, 10–13, *n* = 2, mouse #9, #14, *n* = 1. **b** On-chip drug screening results for the biopsy samples from 14 mice with Dox (40 μM) or EP (40 μM) or Wzb (40 μM) treatment and the corresponding drug administration mode in vivo. **c** fluorescence imaging results for the biopsy samples from mouse 6 with different drugs (Cis, EP, Wzb) treatment. Green color represents live cells, red color represents dead cells. "×", the cell toxicity was not measured due to limited amount of samples. Each experiment was repeated independently for 3 times with similar results. Source data are provided as a Source Data file.

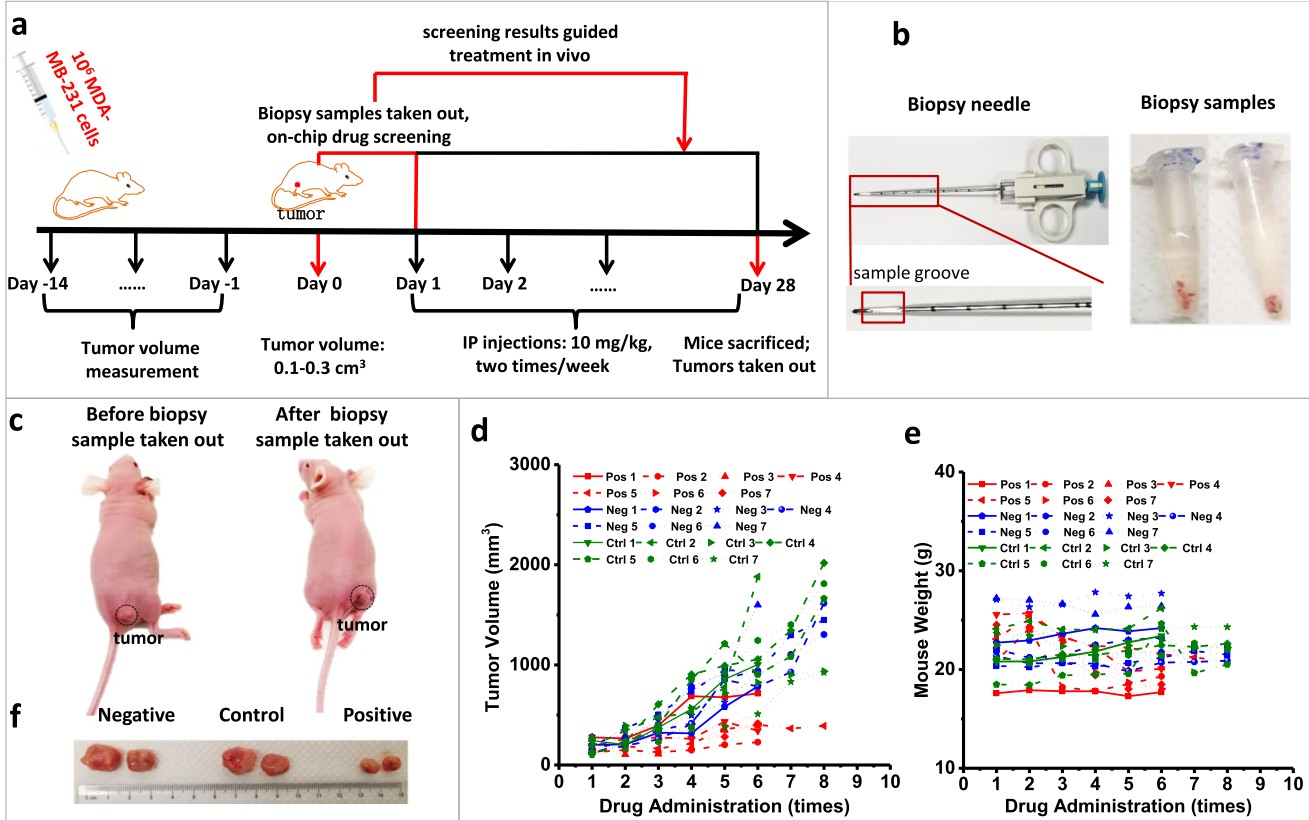

**Fig. 3 | The process and results of single drug therapy in vivo. a** The process involving breast cancer MDA-MB-231 model establishment, biopsy sample collection, on-chip drug screening, and screening results guided treatment in vivo. **b** Image of the biopsy needle and biopsy samples collected from the tumor area. **c** Image of the mouse before and after obtaining a biopsy sample with skin adhesion around the tumor area. **d**–**f** The results of mice treatment. The relationship between drug administration times, **d** tumor volume, and **e** mouse body weight. **f** Tumor size after drug treatment. Source data are provided as a Source Data file.

biopsy sample with skin adhesion around the tumor area. In this study, Cis exhibited a positive effect on mice, while Wzb was ineffective in treating most mice. This could be attributed to the use of the same cell line, MDA-MB-231, to develop tumors in different mice. Although the mice themselves varied, the tumor cells shared the same characteristics, weakening the personalized mouse reaction to drugs. Therefore, Cis was chosen as the drug for the positive group, and Wzb was administered to the negative group.

As shown in Fig. 3d, the tumors of the positive group (treated with the effective drug) were smaller than those of the negative and control groups after one month of treatment, aligning with the on-chip drug screening results. The tumor growth trends within the treatment period were different among mice within the same group; thus, suggesting the existence of individual differences. In the positive group, the tumor volume of mouse #2 was the lowest in the same treatment period, which was consistent with the on-chip drug screening result, where the primary tumor samples from mouse #2 were the most sensitive to the effective drug.

Body weight measurements were used to evaluate drug toxicity. No considerable changes in body weight were observed in the mice from the negative and control groups (Fig. 3e). However, in the positive group, the body weight of the mice was reduced, suggesting the potential toxicity of the effective drug, which aligns with findings reported by Pinzani et al. and Yao et al.[32–34]. This effect has been attributed to nephrotoxicity. Representative images of typical tumors after treatment are presented in Fig. 3f. The tumor volume of the mice in the positive group was lower than that in the other groups, consistent with the in-vitro drug screening indications. These results collectively demonstrate the reliability of the DMF platform for on-chip drug screening to guide in-vivo cancer therapy.

## Combinatorial drug screening of primary tumor samples from MDA-MB-231 breast cancer xenograft mouse model for precision medicine

In the above experiments, single drugs were screened on-chip and administered to mice for investigating precision medicine. Considering the multiple cell types of cancers, which may require a combinatory treatment using several drugs, combinatorial drug screening of primary tumor samples on DMF were further investigated for precision medicine in an MDA-MB-231 breast cancer xenograft mouse model.

In this experiment, Doxorubicin (Dox) and Curcumol (Cur) were used as the drug models. Biopsy samples from 15 mice were used for on-chip drug screening. Guided by the on-chip screening results, mice were sorted into three groups with one group treated with a single effective drug, one group with single least effective drug, and the other group with combinatorial drug. Another group of 5 mice were treated with PBS solution (control). Detailed drug screening results for Dox alone, Cur alone, and the combination of 10 μM Dox and various concentration of Cur are described in the SI (Fig. S4).

Figure 4a summarizes the cell viability at different concentrations of Dox, Cur, and Dox & Cur. Dox alone was more effective than Cur alone in most cases, resulting in lower cell viabilities. The combination of the two drugs was always more effective than the single drugs. Representative on-chip drug screening fluorescence images of the biopsy samples from 3 mice (mice 4, 9, and 13) are shown in Fig. 4b.

To test the correspondence between on-chip drug screening and in-vivo drug therapy, one group of mice was treated with 10 mg/kg Dox as the positive drug group (mice 1–5), one group was treated with 10 mg/kg Cur as the negative drug group (mice 6–10), and one group was treated with the combination of 10 mg/kg Dox and 10 mg/kg Cur as the combinatorial drug group (mice 11–15).

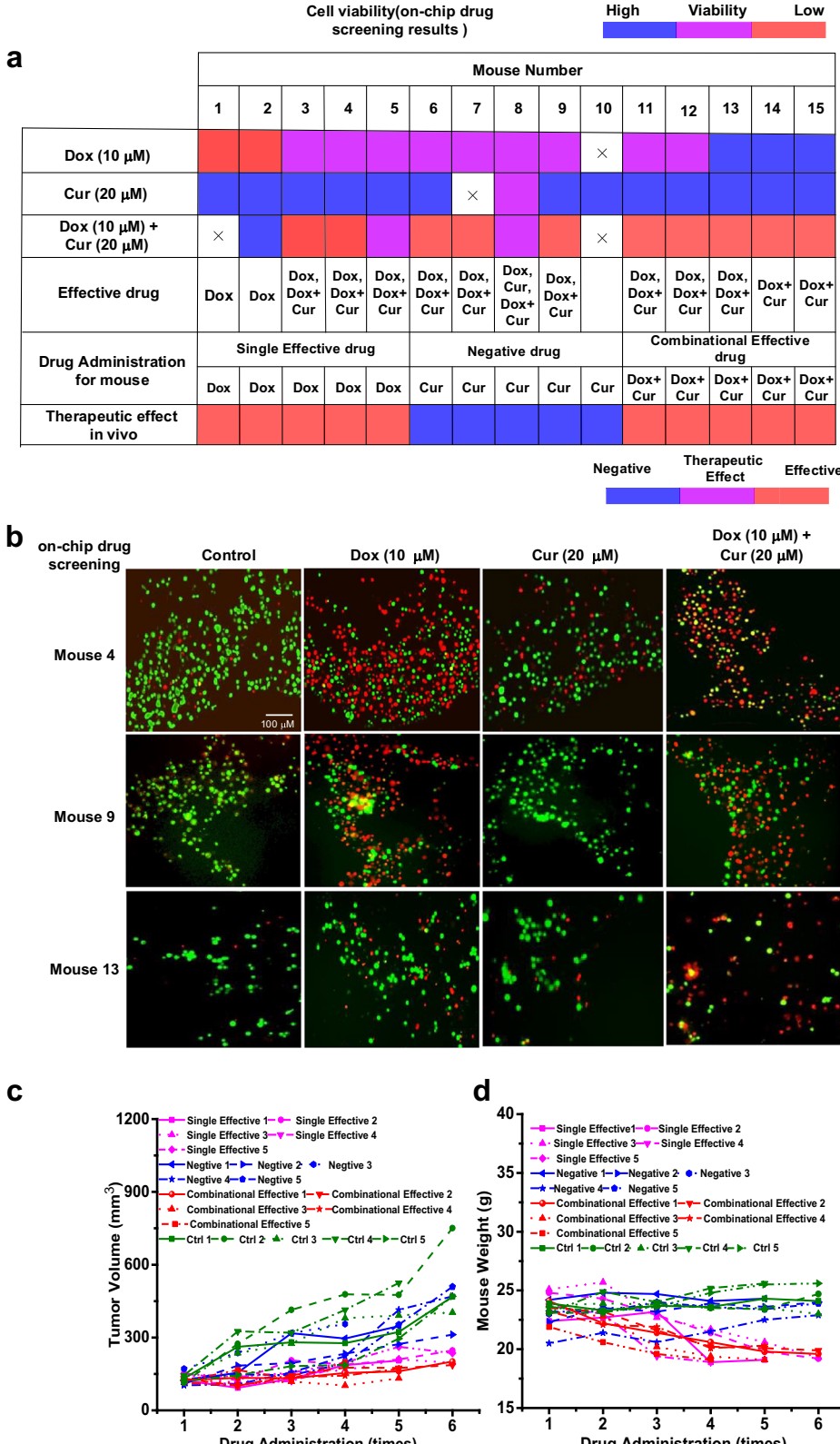

**Fig. 4 | Combinational drug screening on chip and in vivo therapy. a** On-chip drug screening results for the biopsy samples from 15 mice with Doxorubicin (Dox, 10 μM) or Curcumol (cur, 20 μM) or Dox (10 μM) plus Cur (20 μM) treatment and the corresponding drug administration mode and therapeutic effect in vivo. **b** Representative on-chip drug screening fluorescent imaging results for the biopsy samples from mouse 4, 9, and 13 after Dox (10 μM) or Cur (20 μM) or Dox (10 μM) plus Cur (20 μM) treatment. Because of the limited biopsy samples, one experiment was repeated for mouse 4 and mouse 13. Each experiment was repeated independently for 2 times with similar results for mouse 9. **c, d** The results of mice treatment. The relationship between drug administration times and **c** tumor volume, **d** mouse body weight. Source data are provided as a Source Data file.

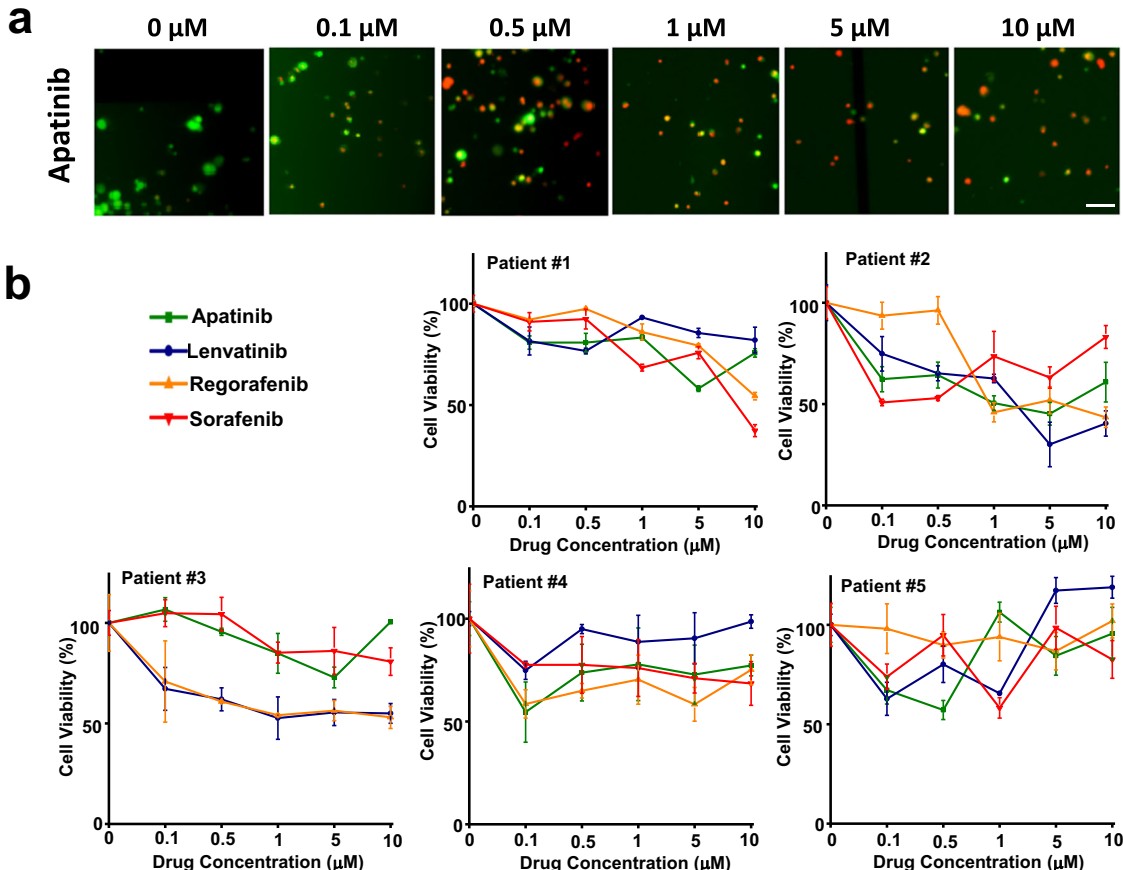

**Fig. 5 | On-chip drug screening results of four clinical anti-cancer drugs on five patients with liver cancer using a portable Digital Microfluidic (DMF) system.** **a** Cell viability results of apatinib-treated group derived from patient #3. Live cells were stained with green fluorescence and dead cells were stained with red fluorescence. Each experiment was repeated independently for three times with similar results. **b** Dose-response results of four commonly used empirical targeted drugs on five patients with liver cancer. Data are the mean of three independent experiments. Source data are provided as a Source Data file.

Figure 4c shows that the tumor volumes of the positive and combinatorial drug groups were lower than those of the negative drug and control groups after one month of treatment, aligning with the on-chip drug screening results. Tumor growth trends within the treatment period were different among mice within the same group, suggesting the existence of individual differences. In the positive drug group, the tumor volumes of mice treated with a combination of Dox and Cur were smaller than those of mice treated with Dox alone. This result is consistent with the on-chip drug screening result, where the combinatorial treatment was suggested to be the most effective.

Body weight measurements were further used to evaluate drug toxicity. As shown in Fig. 4d, no considerable changes in body weight were observed in the mice from the negative drug and control groups. However, in the positive drug group, the body weight of mice was reduced; thus, suggesting the potential toxic effects of drug. These results collectively demonstrate the reliability of the DMF platform for on-chip combinatorial drug screening usage.

### Drug screening for HCC specimens

The MDA-MB-231 breast cancer xenograft mouse model harbored well-developed cancer cell lines that can survive harsh processes, such as in in-vitro on-chip drug screening. Considering the potentially higher fragility of human clinical tumor samples than that of xenograft mouse model, we further assessed the feasibility of the DMF drug screening system using clinical tumor samples for precision medicine. In this experiment, hepatocellular carcinoma (HCC) specimens and four targeted liver cancer drugs were chosen as the cancer and drug model systems, respectively. Owing to the small footprint of the DMF drug screening system, we took the system into the hospital to allow for freshly obtained tumor dissection samples to be dissociated and screened immediately to avoid hypoxia-induced damage to the primary tumor cells.

To minimize potential interferences between the drug screening and cancer treatment results, a double-blinded approach in terms of screening and treatment was employed. All patients were treated following the standard-of-care. The potential targeted liver cancer drugs that could be prescribed to a certain patient were screened on-chip. At the same time, the patient was treated empirically by a doctor unaware of the drug screening results. Patients who received the screened effective drugs were sorted into the positive group, while those who received ineffective drugs were sorted into the negative group. Patients not treated with any drug were included in the control group. A drug was considered effective if the relative cell viability reached 50% at the highest drug concentration. Since the tumors had been dissected, tumor recurrence was used as a marker to determine treatment efficacy. Four commonly used targeted liver cancer drugs, namely, Lenvatinib (Len), regorafenib (Reg), apatinib (Apa), and sorafenib (Sor), were screened on-chip using five clinical HCC specimens. Cells were stained with Cell Tracker™ Green CMFDA Dye and EthD-1 to label the live and dead cells with green and red fluorescence, respectively, and observed after 24 h of culturing. Figure 5a displays representative images of primary tumor cells from patient #3 treated with Apa. The cells maintained their morphology at all drug concentrations. When no drug was administered, almost all cells were alive. This suggests that the drug screening process did not damage the primary tumor cells, validating the indications of cell responses to drugs.

**Table 1 | Targets and mutational profiles of the three patients**

| Genes | Target drugs | | | | Patient #1 | Patient #3 | Patient #5 |
|---|---|---|---|---|---|---|---|
| | Apatinib | Lenvatinib | Regorafenib | Sorafenib | | | |
| TP53 | | | | | + | − | + |
| ALK | | | | | + | − | − |
| NF1 | | | | | + | − | − |
| MET | | | | | − | +↑ | − |
| RET | √ | √ | √ | √ | − | − | − |
| FGFR1 | | √ | √ | √ | − | − | − |
| FGFR2 | | √ | √ | | − | − | − |
| FGFR3 | | √ | | | − | − | − |
| FGFR4 | | √ | | | − | − | − |
| BRAF | | | √ | √ | − | − | − |
| PDGFR-α | | √ | √ | | − | − | − |
| KIT | √ | √ | √ | √ | − | − | − |

"√" drug targeting genes, "+" mutated, "-" no mutation, "↑" up-regulated expression.

**Table 2 | HCC patients' clinical information[a]**

| Patient # | Sex | Age | Immuno-suppressed | Metastasis | In-vitro effective drugs | In-vivo treated drugs | Group | Cancer recurrence |
|---|---|---|---|---|---|---|---|---|
| 1 | Male | 51 | N | N | – | – | Ctrl | N |
| 2 | Male | 69 | Y | Y | Lenvatinib | Lenvatinib | Pos | N |
| 3 | Male | 40 | N | N | – | – | Ctrl | N |
| 4 | Male | 69 | N | N | – | Corida + lenvatinib | – | – |
| 5 | Male | 67 | N | N | – | Atenolizumb + bevacizumab | – | – |

[a]Cancer recurrence: patients were followed up in 6 months.

Higher concentrations of Apa resulted in increased cell death, as expected. However, for an Apa concentration of 10 μM, the cell viability remained above 50% for patient #3. Concentrations higher than 10 μM were not tested due to the anticipated toxicity that would limit the realistic prescription dosage for patients[35]. Figure 5b shows the viability of primary tumor cells from each patient in response to the four drugs. The primary tumor cells of each individual patient had a different response to a certain drug, consistent with our findings in mice. Patient #1 responded moderately to Reg and Sor but minimally to Apa and Len. Patient #2 demonstrated a favorable response to Len, with 50% cell viability at approximately 2 μM and a consistent decrease in cell viability at higher concentrations. Cells treated with Reg hardly reached 50% viability at certain concentrations. Other drugs showed an inconsistent effectiveness in samples from patient #2. For patient #3, Apa and Sor did not show an effect at any concentration, whereas Len and Reg led to reduced cell viability at high concentrations. However, none of the samples reached 50% viability. The primary tumor cells of patients #4 and #5 showed no responses to any drug. Especially for patient #5, Len even increased cell viability at high concentrations, indicating a favorable environment for cell growth in the presence of Len. In summary, Len was identified as potentially effective in patient #2, while no effective drug was identified for the other patients.

To confirm the drug efficacy in each patient and determine why the patients had such low responses to the four targeted drugs, the remaining samples were sent out for whole exon sequencing covering common carcinoma-related genes and targeted genes related to these four drugs. Owing to the limited number of tumor samples from patients #2 and #4, sequencing data could not be obtained. The gene mutation results for patients #1, #3, and #5 are presented in Table 1 (exactly mutation for these patients are shown in Fig. S5). The genes related to the four target drugs (Apa, Len, Reg, and Sor), including *RET*, *FGFR1*, *FGFR2*, *FGFR#*, *FGRF4*, *BRAF*, *PDGFR-α*, and *KIT*, were assessed.

As shown in Table 1, no mutations were identified in any genes related to the four drugs in patients #1, #3, and #5. This finding was consistent with our in-vitro drug screening results in that none of the three patients had apparent responses to the tested drugs, validating the drug screening results at the genetic level. We also observed that patients #1 and #5 had mutations in the oncogene *TP53*, which encodes for a protein located inside the cell nucleus and plays a key role in controlling cell division and death. Mutations in *TP53* may lead to uncontrolled cancer cell growth and spread in the body[36]. Especially for patient #1, mutations were also identified in *ALK* and *NF1*, which encode proteins involved in cell growth. The mutated forms of these genes and proteins also promote uncontrolled cell growth and lead to various cancers[35,37,38]. This may explain why patient #1 developed cancer at a rather young age of 51 years (Table 2). Patient #3 harbored a mutation in *MET*, resulting in increased *MET* expression. *MET* is a proto-oncogene that encodes a tyrosine receptor kinase protein involved in cell growth and survival. The overexpressed MET has been shown to interact with the vascular endothelial growth factor (VEGF) and VEGF receptor (VEGFR) pathways, promoting angiogenesis and endothelial cell growth. Angiogenesis is fundamental for tumor growth, invasion, and metastasis[39]. The molecules interfering with blood vessel formation have been shown to block tumor progression. VEGF plays an important role in tumor angiogenesis[40,41]. Currently, all approved therapies for HCC are molecular-targeted therapies with anti-angiogenic effects. The primary mechanism underlying anti-angiogenesis is to target VEGF and its receptors[42]. As VEGFR is also a target of Len[43] and Reg[44], albeit not a main one, mutations in *MET* may make these two drugs relatively effective. This indication was consistent with our drug screening results for patient #3.

As analyzed above, the in-vitro drug screening results were consistent with the data from exon gene sequencing, suggesting the feasibility of using primary tumor cell-based drug screening as a cost-effective alternative to whole gene sequencing for precision medicine.

To test the effectiveness of precision medicine based on the in-vitro drug screening, we compared the drug screening results and in-vivo clinical drug treatment results in a double-blinded manner, as presented in Table 2. Histological image and CT imaging were performed for observing the recurrence of the tumor to assess whether the drug was effective in vivo (Figs. S6 and S7). As presented in Table 2, patients #4 and #5 were not sorted into any group as drugs not screened on-chip were prescribed to them. Patients #1 and #3 were not treated with any drugs and were therefore sorted into the control group. Patient #2, who exhibited a positive response to Lenvatinib in the in-vitro drug screening, was assigned to the positive group. After 6 months of follow-up observation, patients #1–3 exhibited no tumor recurrence. Notably, a tumor is resected when it can be completely removed with safe margins, leading to a low probability of recurrence. In case of a completely resected tumor, the tumor recurs generally 2–5 years after surgery. The cancer-free status of the two control group patients may be due to the limited follow-up time after complete tumor resection. For patients (such as patient #2) that received liver transplantation owing to liver cancer before this surgery, the tumor was suspected to emerge from remaining original cancer cells in the patient. When only one tumor is noted at a location and it can be completely resected, surgical resection remains the first line of treatment. However, the chance for cancer recurrence in these cases is much higher than that of a sole early stage tumor. Given the situation of the patient, doctors decided to treat him with a targeted therapy after surgery to clear up the remaining cancer cells. The targeted therapy used for this patient happened to be the effective drug according to the on-chip drug screening, rendering him in the positive group. Given the high chance of this patient to experience cancer recurrence, the cancer-free status for the past 6 months might be mostly due to the drug treatment rather than the tumor resection. Therefore, the lack of cancer recurrence in patient #2 indicated the effectiveness of the screened drug.

To further assess the clinical liver cancer biopsies, we performed flow cytometry analyses. In this experiment, the combination of CD44 and CD24 were used to identify the existence of cancer stem cell (CSC)[45]. Cells with high CD44 expression and low CD24 expression were considered CSCs. A high expression is indicated by the right shift in the intensity peak of stained cells compared to that of unstained cells. As shown in Fig. 6a, b, the CD44 peaks of the stained cells from two biopsy samples were right shifted compared to those of the unstained cells. The CD24 peak positions of both the samples had no obvious difference for stained and unstained cells, indicating a low CD24 expression. The high expression of CD44 and low expression of CD24 demonstrated the presence of CSCs in the biopsy samples.

Furthermore, we examined the CD34, CD45, CD14, CD19, and HLA-DR expression patterns, which are associated with hematopoietic and immune cells[46]. As shown in Fig. 6a, b, the intensity peaks were similar for all markers and there were no differences between stained and unstained cells. This indicates the presence of a low number of immune cells in the biopsy sample. Based on these data, we concluded that the drug screening of the biopsy sample mostly reflected the tumor cell responses to the screened drugs.

The cell viability in these samples was also measured to demonstrate their in-vitro growth potential for drug screening. As shown in Fig. 6c, d over 80% cell viability was observed in the three samples. This finding further indicates the reliability of the drug screening on-chip results.

## Discussion

Gene expression assessments have been at the forefront of the advancements in precision medicine. Such assessments account for individual variability in genes, environment, and lifestyle for each patient. However, whole gene sequencing, even only one exon sequencing, costs thousands of dollars for one sample. Furthermore, prescribing medication solely on gene sequencing data is not reliable due to the various pathways in which proteins can be involved. Approximately 15–30% of patients showed unrelated responses to a potential drug that was prescribed based on gene sequencing[47,48]. This may be attributed to the number of unknown genes which are also related to this drug, but not sequenced. To overcome this issue, we propose the use of primary tumor cells drug screening as the indication of drug effectiveness. Regardless of the mutations harbored by the cells, the drug responses of primary tumor cells always reflect the direct responses of the tumor itself. However, traditional drug screening based on cell culture requires millions of cells, this being a bottleneck for applying it on primary tumor cells[49–51].

To work with a limited number of primary tumor cells, in this work we developed a DMF device that can be taken into animal facilities or hospitals for immediate operation on fresh samples. This minimizes hypoxia- or temperature drop-induced cell death. The DMF system has several advantages: (1) Compared to traditional DMF chips which use one control pin for one electrode control, the smart electrode sharing scheme increases the drug screening throughput. With a traditional control system, 24 actuation signals can only operate 24 electrodes, while the sharing scheme in this study allows 24 actuation signals to control 96 electrodes. More drugs could be screened on-chip with the same number control signals. (2) The small footprint increases the portability of the device. (3) The entire workflow from primary tumor sample collection and to drug screening results can be completed in 36 h. The number of primary tumor cells is usually too low for traditional drug screening in a microplate. Culturing the primary cells is required to generate enough cells for drug screening, which is time consuming. With our system, the primary tumor cells were directly screened on-chip. The entire workflow can be completed in 36 h. The entire workflow includes tumor sample acquisition after surgery, cell dissociation, on-chip droplet operation, on-chip cell culture for drug screening, and data analysis, which in total takes around 36 hours. The cells experience hypoxia only in the first step, where the tumor has been dissected from a patient but not dissociated yet. Cell dissociation and cell culture are all in the incubator. Cell damage can be observed after 6 hours if the sample was left untreated after surgery (Fig. S8 a-c), so the fresher the tumor samples, the better the results. This is also why we built the device as a portable one that can be taken to animal facilities or hospitals to start experiments using the freshest samples. More importantly, sub-culturing primary cells may introduce unexpected mutations in descendant cells and affect the drug screening results. (4) The cost can be reduced to less than one hundred dollars for each sample. The net cost of the microplate ($1.5 for a 96-well plate) is low compared to that of the DMF chip ($20 per chip). However, drugs are expensive. For example, Len costs $140 and EP costs $160 for 10 mg. To maintain the same drug concentration in a 96-well plate with 100 μL solution would cost 300 times more than the costs associated with the use of the DMF chip with 0.3 μL solution. Furthermore, the DMF chip can be washed and surface treated again for reuse. If one DMF chip can be reused 10 times, the cost per drug condition would be 3000 times lower than that associated with the use of microplates. (5) The capability to work with primary tumor cells eliminates the need for cell passage cultivation before drug screening, and therefore providing the direct information of drug sensitivity from the tumor for precision medicine.

To validate our DMF drug screening system for precision medicine, we tested it on an MDA-MB-231 breast cancer xenograft mouse model. We have validated it with different batches of mice, different

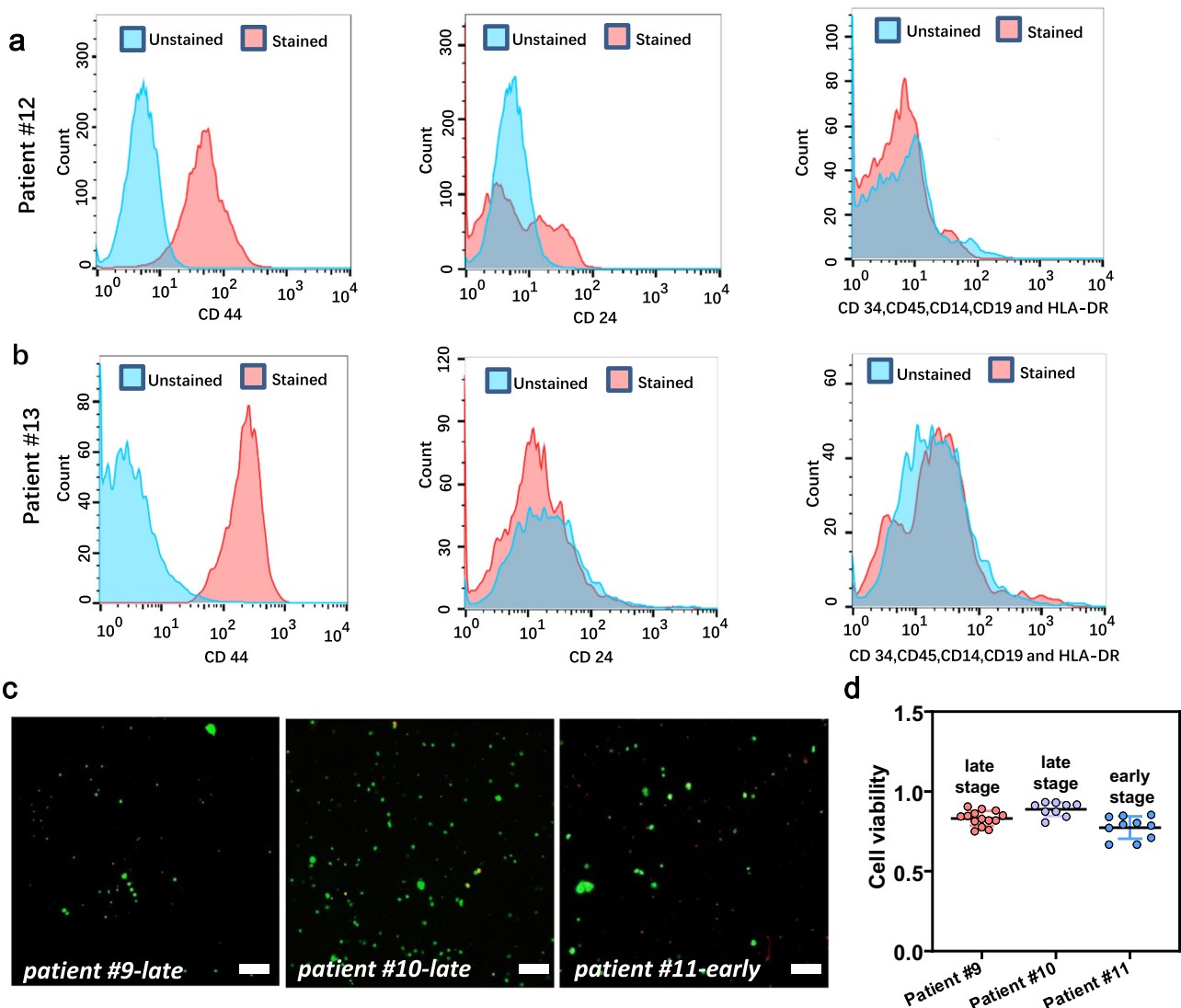

**Fig. 6 | Characterization of the biopsy samples from human patients. a, b** Flow cytometry analysis of patients#12, #13, revealed cells with high CD44 but low CD24 expression. **c** Fluorescence imaging results of the cells obtained from biopsies of patient #9, #10, and #11. Green represents living cells and red represents dead cells. Scale bars are 100 μm. **d** Cell viability results of the samples from patients #9 ($0.83 \pm 0.047$, $n = 14$), #10 ($0.89 \pm 0.04$, $n = 9$), and #11 ($0.77 \pm 0.07$, $n = 10$). Source data are provided as a Source Data file.

types of drugs and drug combinations under strict controls. The in-vitro drug screening results matched well with the in-vivo drug treatment on individual mice for tumor suppression. To further validate the system for clinical samples, we compared the in-vitro drug screening results for four targeted drugs in 5 liver cancer samples with exon sequencing data. The ineffective drugs in certain patients were consistent with the absence of mutations in their related genes. We further compared the in-vitro drug screening results with in-vivo personal treatment in a double-blind manner. The positive drug successfully delayed the recurrence of a metastatic tumor in one patient; that is, no recurrence was observed after six months. This further validates the potential of precision medicine based on the DMF drug screening system. More critically, unlike the proof-of-principle demonstration of the DMF functions in various applications in literature, the DMF device presented in this paper is a complete integrated portable equipment that functions in animal facilities and hospitals. None of the existing channel-based microfluidic systems has achieved this. The sample size for the drug treatment experiment was small due to the double-blinded experiment design, which would leave some space for

uncertainty. In the future, we will use more samples and more types of cancers for validation.

As with any newly emerging technique, the current drug screening system has limitations.

The control system is not totally automatic, although previous studies introduced the automatic control of droplet movement[52–58]. In the automatic controls, either a set of electrode charging steps were programmed for completing a series of steps which lacks error-tolerance, or another screening signal was provided in addition to the actuation signal for real-time location of a droplet which adds another level of electronic signal. For primary tumor cells, which are more vulnerable than commercialized cancer cells, simple controls are preferred to minimize the influence of electric signal on the primary tumor cells. However, without real-time droplet monitoring, a program that worked for one drug may fail for another drug due to differences in viscosity, hydrophobicity, or surface tension value. After considering all factors, we chose a push-down button design, allowing us to monitor where the drop was and adjusting the actuation to make sure all the droplets went to the correct location. This allows us to

preserve most sample cells. However, the push-down button strategy is only temporary. We are currently developing a more reliable automatic control system with on-chip drug delivery. We expect that it will make the control safe, convenient and robust in the future. In this study, we focus on validating the DMF screening system of primary tumor cells and prove that it provides reliable instructions for precision medicine.

The cell number used on DMF chip for drug screening did not reach the maximum limit. As shown in Fig. S3a, b approximately $5 \times 10^4$ cells were obtained in a biopsy sample. During screening and preparation, some cells may be lost to centrifugation or half-filled biopsy needles. Therefore, roughly 500 to 1000 cells per dose, and 3000 to 6000 cells per drug were used for screening on-chip. Eduati et al.[9] used approximately 100 live cells for screening each drug dosage, contained within a 500 nL droplet, as part of a channel microfluidics setup. We predict that we could produce reliable results using the DMF system and a similar number of cells per droplet. We could achieve this by adjusting the size of electrodes on the DMF chip to reduce the size of the droplet to that of the droplet used in the channel microfluidics setup.

Primary tumors were cultured in 2D, neglecting the microenvironment in vivo[59]. 3D culture is the foremost technique and the final target of our system development for drug screening on primary tumor cells. However, current 3D organoid models still cannot completely recapitulate the dynamic tumor environment including fibroblasts, endothelial cells, immune cells, and extracellular matrix[60,61]. The effectiveness of 3D organoid models in drug screening for precision medicine is inconsistent, with success rates ranging from 15% to 90%, leading to uncertainty in their application[62,63]. A potential applicable 3D culture strategy on a DMF chip could use Matrigel to solidify a droplet on-chip for primary tumor spheroid growth[64–66]. Another approach to observe the primary tumor cell responses to drug involved the use of a thin slice of tumor which already has the same microenvironment as an in-vivo tumor[59,67]. We will explore these approaches in the future studies to improve our drug screening system for 3D primary tumor drug screening. A higher throughput would also be a way to further optimize the system for the mass production of the chip and mass operation of a large number of samples. These efforts would promote advances in precision medicine for cancer treatment and eventually for any disease treatment.

## Methods

### System setup

The portable DMF device (Fig. S9) contained four parts: a two-layer electronic control board, a DMF chip, a connector to connect the electronic control board and DMF chip, and a 5 V power adapter. A 3D-printed chip holder was used to hold the DMF chip. Fluorescent microscopy was used to observe the DMF chip (Fig. S10).

The electronic control board consisted of mechanical buttons arranged in the same pattern as the electrodes on the DMF chip, facilitating clear indication of droplet movement. Additional functional buttons were arranged at the top of the device to adjust the electric signal parameters for an optimized signal under various conditions. The signal generator screen displayed the real-time electric voltage, providing information on the real-time actuation signal. The real-time actuation signal is only to provide additional information on how much voltage was applied to actuate the droplet with a certain drug, so that similar voltage can be set directly for the same drug. It is not mandatory request for operation.

The functional buttons on the signal generator enabled adjustment of the actuation signal frequency, voltage, and waveform to provide an appropriate actuation signal, which was then amplified by an electromagnetic transformer to a droplet actuation signal ranging from 70 V–120 V. Multiple paths of actuation signal were connected to

the contacting pad on the DMF chip using jumping wires to drive droplet transportation. A low voltage through the button switch is desired for safety considerations in portable applications. Hence, physical relays, capable of switching on/off an electric circuit with a 5 V power supply, were used as the switches. When a switch button was pressed, a current from the power adapter with a 5 V power supply passed through the physical relay, switching on the AC circuit to supply the 100 V voltage actuation signal to the corresponding electrodes on the DMF chip. To fit a portable-sized device, a total of 24 physical relays were integrated into the PCB to control 24 lines of signal input. The outlook of the portable DMF drug screening device is shown in Fig. S1a.

The DMF chip design for multiple-drug screening is shown in Fig. S2a. Three identical patterns of electrodes were arranged on the chip, each designed for screening one drug. The pattern was the same as that on the control panel, enabling easy control of the droplets and the potential for tailored droplet movements. Fig. S2b showed the dynamic process of on-chip drug screening. We actually used automatic control for droplet transportation when starting the project. But, we found out that a program that worked for one drug may fail for another. We suspected it was because of the diverse properties of different drugs. We made a step backward for push-button control, so that we can change the actuation voltage, actuation time, and the droplet movement path to make sure each drop can safely reach the position it should go.

Although only 24 lines of signal input were connected to the control pads lining the two edges of the DMF chip, 96 electrodes were actuated for multiple-drug screening. Supplementary Movie 1 demonstrates the parallel drugs screening on a DMF chip. To achieve effective individual droplet control, electrode-sharing logic was employed, connecting multiple electrodes to the same control line. All electrodes marked with the same number were connected to each other and to one line of the input voltage. Supplementary Movie 2 shows three droplets moving simultaneously based on the electrode-sharing principle. The algorithm for sharing electrodes is described in detail in the next section.

### Smart electrode-sharing protocol for drug screening

Conventional DMF biochips employ *direct addressing*, requiring one control pin for each electrode on the chip. With an increase in the number of electrodes, the number of control pins and routing complexity increase significantly. The direct addressing scheme can become impractical owing to the increased cost of the control pins. To address this, *broadcast addressing* is introduced to reduce the required number of control pins by allowing each control pin to address multiple electrodes without impacting the scheduled droplet movements. However, for different problem sizes corresponding to different chip sizes, wire routing and the control logic must be recomputed, leading to significant increases in the design and control costs. To enhance the suitability of the DMF chip for high-throughput drug screening, we propose a smart electrode-sharing protocol for drug screening featuring a novel electrode connection structure and control algorithm.

By interconnecting repeatable units in series, the electrodes located at the same position as the repeatable unit can be routed to a single control pin without affecting the designated functionality. With this protocol, the number of control electrodes was fixed at ten, regardless of the number of storage units. The wire routing scheme of the chip was fixed, and the control logic of the chip was straightforward.

Before introducing the proposed smart electrode-sharing protocol, we first provide the related definitions as follows:

Definition 1: The *activation state* is determined by the control signal from the control pin. Three activation states exist: (1) "0" denotes an inactivated state with low voltage, (2) "1" denotes an

activated state with high voltage, and (3) "X" signifies that the electrode can be set to any state (activated/inactivated).

Definition 2: A *time step* is defined as the unit of time necessary for moving one droplet. At each time step, a droplet on an electrode has three choices: (1) move forward onto an adjacent electrode, (2) move backward onto the previous electrode, and (3) stall at its current position.

Definition 3: An *activation sequence* for electrode $e_i$ is defined as a sequence of activation states $(a_{i,1}, a_{i,2}, ..., a_{i,n})$, where $a_{i,k}$ ($1 \le k \le n$) can be assigned "0," "1," or "X." Here, $n$ is the total number of time steps for executing the drug screening protocol.

The proposed chip structure is illustrated in Fig. S1c and consists of three areas: the input, storage, and control areas. The storage area featured a fishbone structure comprising multiple repeatable units, with each unit containing five electrodes and two storage units. The wire routing within the repeatable unit followed a fixed pattern, eliminating the need to recompute the wire-routing solution as the number of storage units increased. The number of repeatable units was computed by dividing the number of storage units by 2. The control area comprised seven control pins that addressed the seven electrodes contained in the repeatable unit. The input area included three inlets and nine electrodes. Drug droplets are introduced through the inlets and guided along electrodes toward the junction of the input and storage areas. To reduce the number of control pins, three electrodes from the top inlet and three electrodes from the bottom inlet share three control pins (10, 9, and 8). The remaining five electrodes in the input area shared control pins (5, 3, and 4) with electrodes in the storage area. Thus, only three additional control pins were required in the input area. Notably, the ten control pins in our proposed chip structure remained unchanged regardless of the number of storage units.

According to the proposed biochip structure depicted in Fig. S3c, activating the three pins (5, 3, and 4) in sequence can facilitate droplets to move three steps forward along the spine of the fishbone. Thus, repeated activations of these pins (5, 3, and 4) caused the droplet to move continuously along the fishbone until it reached the repeatable unit, where the target storage unit was located. When the droplet was to be moved to the upper/lower storage unit in the $k^{th}$ repeatable unit, the corresponding activation sequence of the control pins was "10, 9, 8, (5, 3, 4)$^{(k+1)}$, ((2, 1)/(6, 7))." In each time step, only one control pin was set to a high voltage, and the other control pins were grounded. For example, if a droplet was to be moved to the upper storage unit in the first repeatable unit, the activation sequence of the control pins was "10, 9, 8, 5, 3, 4, 5, 3, 4, 2, 1." The corresponding activation sequence and chip status at different time steps are shown in Fig. S1d.

To maximize the controlled droplet using a limited number of control signals, Yang et al. and Perry et al. introduced "modular bussing strategies"[68,69]. The structure proposed by Yang[68] can only be used to control the two-way movement of the droplets. To make the DMF chip more suitable for high-throughput drug screening, we extended it perpendicular to the bus. The extended repeating unit not only guarantees the fixed number of control electrodes, but also ensures the minimization of the total number of electrodes required to achieve the miniaturization and portability of the entire drug screening system. Based on the chip structure we proposed, we further designed a templated control logic scheme for the valuable drug screening task, which can determine the control logic of drug screening in a constant time. The proposed method avoids the overhead using complex algorithms to calculate the control logic and ensure the correctness of the drug screening function. Perry[69] investigated repetitive electrode units but not signal sharing between electrodes. Therefore, the number of control pins increased proportionally with the number of repeating units. In this study, to further minimize the control pins, the repeatable units also include a unique wiring structure which ensures that they are connected to the same control. The number of control pins is fixed even when the number of repeat units increases.

## DMF device fabrication

The DMF chip consisted of three parts: the bottom plate, spacer, and top plate. Three parallel electrode pattern array groups were designed using AutoCAD and patterned on a glass substrate (40 mm × 40 mm) as the bottom plate.

About using other substrates such as PCB for convenient and powerful controllability like active matrix control[70], we actually found out glass substrate is the most appropriate material. For example, the surface flatness of PCB does not meet the requirement of single cell observations. As shown in Fig. S11 (a, b), the scratch on the copper electrodes would cover the signal of cells, making it difficult for cell identification. LCD could be another option with active matrix control[71], but the cost has limited its application for biological samples. So, we chose a glass substrate and combined it with the electrode-sharing algorithm to achieve the same control ability with fewer control signals.

For the chip fabrication, SU-8 was coated on the bottom plate at a thickness of 10 µm as the dielectric layer. After development, a second SU-8 patterned layer (60 µm in thickness) was coated as a fence to prevent the droplets from drifting[72,73]. A mask aligner (ABM, California, USA) was used for precise patterning of the coating layer during fabrication. ITO glass was cut to the dimensions of 35 mm × 20 mm as the top plate. A laser cutting machine (ZKJ Laser, Shang Hai) was used to drill sample-loading holes into the ITO glass. Both the top and bottom plates were coated with Teflon of 100 nm in thickness to promote smooth droplet transport. Conductive adhesive tape with a thickness of 100 µm was used as a spacer.

## Reagents

Acetone, ethanol, and IPA were purchased from Millipore. The SU-8 and SU-8 developers were purchased from MicroChem. The amorphous fluoroplastic solution was purchased from Chemours Company. Pluronic F127 was purchased from Sigma-Aldrich (Oakville, ON, USA). Silicone oil (1 Cst) was purchased from Clearco, USA. Fetal bovine serum (FBS), PBS, collagenase II, Hank's Balanced Salt Solution (HBSS), Earle's balanced salt solution (EBSS), DMEM/F12, Glutamax, HEPES, and 1:50 B27 were purchased from Gibco. Cis-diammineplatinum (II) dichloride, EP, Y27632, dexamethasone, penicillin/streptomycin, N-acetyl-l-cysteine, nicotinamide, insulin, hydrocortisone, cholera toxin, and hyaluronidase were purchased from Sigma. Wzb117 was purchased from Selleckchem. Recombinant human EGF, recombinant human FGF10, and recombinant human HGF were purchased from Peprotech. RBC lysis buffer, EthD-1, erythrocyte lysate, and Cell Tracker™ Green CMFDA Dye were purchased from Invitrogen. StemMACS iPS-Brew XF medium was purchased from Miltenyl Biotec (USA). Dimethyl sulfoxide, Sor, Reg, Apa, Len, and DNase I were purchased from Solarbio. Forskolin and A8301 were purchased from Tocris.

## Droplet manipulation on DMF chip

For on-chip droplet manipulation, two inlets were designed in each group of electrodes: one for the input of cell samples and the other for loading a drug at various concentrations. Fig. S2b schematically shows the loading process of the drug and cell samples on the chip. The two droplets were transported to the common path, mixed, and moved to the culture electrode at the end of the path. Drugs were loaded in a serial order from low to high concentrations to avoid cross-contamination, as low concentrations had minimal effect on the higher ones. Cell culture spots were pretreated with gelation to promote cell adhesion and growth. A 2.5 µL pipette was used to dip the gelation solution onto the cell culture electrodes and let it air dry during the chip fabrication process. Actually, in our later experiments with clinical samples, we found out the gelatin coating was not a must step for the

successful cell culture and drug screening. However, since some experiments were gelatin treated, we described the real experimental conditions in the mouse experiments. It is not a mandatory condition.

The three patterns could be run in parallel, enabling simultaneous loading and culturing under automatic control. Once all the samples were loaded, the chip was put in an incubator for 24 hours before taking out for cell counting of live and dead cells for calculation of cell viability to indicate the efficacy of a certain drug at a certain concentration. The droplet and cell droplet actuation on the DMF chip is demonstrated in Supplementary Movie 3 and 4. The droplet can be collected from the culture spot after screening for further analysis, as shown in Supplementary Movie 5.

### Xenograft nude mouse models

All animal experiments were performed in accordance with the Macau Animal Welfare Act. Human breast cancer MDA-MB-231 cell suspensions ($2 \times 10^6$ cells) (100 μl) were injected subcutaneously into the right flank of individual female nude mice. Sixty-five mice were used in the experiments and labeled with numbered ear tags. The mice were 6 weeks old. During the study, the mice were monitored every alternate day. When the tumor was palpable, a caliper was used to measure the tumor size in two dimensions (length and width). Tumor volumes were calculated using the formula $a \times b^2/2$ ($a$ and $b$ represent the longest and shortest diameters, respectively)[74]. When the tumor volume of the mice increased to 0.1–0.3 cm³, the mice were anesthetized using avertin (250 mg/kg), and a core biopsy needle (16 G × 9 cm, 11 mm sample groove) was used to remove the primary tumor samples from the mice. Each biopsy sample contained ~10,000 cells. Notably, the tumor growth speed of each mouse differed due to mouse-to-mouse variations; the mice with tumors >0.3 cm³ or <0.1 cm³ were discarded in the experiment (the mice with tumors >300 mm³ (454.9496 mm³, 365.3926 mm³, 342.4013 mm³) or <100 mm³ (40.768 mm³, 89.1015 mm³, 85.0023 mm³, 97.3814 mm³, 75.504 mm³, and 49.3293 mm³, were not used for the experiments.). We chose this tumor volume range (0.1–0.3 cm³) for the following reason: A tumor <0.1 cm³ is too small to fully fill in the biopsy needle grove. The number of tumor cells would not be enough for three drug screenings. A tumor >0.3 cm³ may grow too big if the follow-up drug treatment is inefficient. Mice with tumor >2 cm³ must be mercy killed according to our university ethics rules. They may not last the whole observation period we designed. A total of 21 mice and 20 mice were used for sample collection for the single and combinational drug screening research. The obtained biopsy sample pieces were put into 1.5-ml sterile centrifuge tubes with PBS and labeled with numbers corresponding to the ear tags. Then, the tiny skin wound around the tumor of the mice was sealed with medical glue to prevent infection. The biopsy samples obtained were used for primary tumor dissociation and on-chip drug screening.

### Primary tumor dissociation from mice

We first characterized the information of the obtained biopsy from two mice. The HE staining results from Fig. S12 suggested almost all of the cells were tumor cells, with the evidence of the similar morphology of cells in the slides and big nuclear to cytoplasmic ratio of the cells. Primary tumor samples were dissociated as described previously[75] and simplified. Briefly, the mouse biopsy sample pieces obtained using the biopsy needle were first transferred to individually labeled 24-well plates and then washed with PBS twice. After discarding PBS with a pipette, 0.5 ml of Digestion Buffer I (DMEM/F12 medium containing 5% FBS, 5 μg/ml insulin, 500 ng/ml hydrocortisone, 10 ng/ml epidermal growth factor (EGF), 20 ng/ml cholera toxin, 300 U/ml collagenase III, and 100 U/ml hyaluronidase) was added to the wells. Then, the plates were placed into a humidified incubator (37 °C, 5% CO₂) and shaken at 100 rpm for digestion for approximately 3 h. The solution was pipetted every 30 min to accelerate dissociation. Then, the suspensions in each

well were individually pipetted into 1.5-ml sterile centrifuge tubes and spun down at $400 \times g$ for 3 min. After that, the supernatant in the tube was discarded, 0.5 ml of RBC lysis buffer (eBioscience, USA) was added to the tube for red blood cell lysis for 30 s, and finally, 0.5 ml of HBSS (Life Technologies, USA) was added to stop the lysis. Finally, the cells were counted via cytometry, resuspended in StemMACS iPS-Brew XF medium (Miltenyl Biotec, USA) at a cell density of $1.5 \times 10^6$ cells/ml in PCR tubes, and used for on-chip drug screening.

### Drug screening of primary tumor cells from MDA-MB-231 breast cancer xenograft mouse on-chip

For the single drug screening, using Cis, Wzb (glucose transporter 1 inhibitor), and EP as drug models, we monitored their toxic effect on the dissociated primary biopsy tumor cells on our chip over 24 h. First, a series of concentrations (0, 2, 10, 20, 40, and 80 μM) of Cis, Wzb, and EP were prepared via serial dilution with pipette off-chip. Pluronic F127, Cell Tracker™ Green CMFDA Dye and EthD-1 were then added to the dissociated primary tumor cell suspensions of each mouse ($1.5 \times 10^6$ cells/ml) in tubes at a final concentration of 0.01% and 2 μM, respectively. The DMF chip was then filled with silicone oil (1 Cst). Cell suspensions and the drugs (Cis, Wzb, EP) at a series of concentrations (0, 2, 10, 20, 40, and 80 μM) were loaded on-chip through inlet holes, moved under an actuation voltage to the neighboring electrodes sequentially toward the target cell culture electrodes, and mixed on-chip. The chips were then placed in an incubator for 24 h. Finally, cell Tracker™ Green CMFDA Dye entered live cells and emitted green fluorescence. EthD-1 entered dead cells and emitted red fluorescence. Red fluorescent and blue fluorescent images were captured under 10× magnification using inverted fluorescent microscopy (Olympus). The absolute cell viability was calculated as the number of green cells/total number of both green and red cells. The relative cell viability under various drug treatments was normalized to the cell viability without the addition of drugs.

For combinational drug screening, using Doxorubicin (Dox),Curcumol(Cur) and Doxorubicin (Dox) plus Curcumol(Cur) as drug models, we monitored their toxic effect on the dissociated primary biopsy tumor cells on our chip over 24 h. First, a series of concentrations (0, 2, 10, 20, 40, and 80 μM) of Dox, Cur, the mixture of Dox and Cur (0, Dox 20 μM, Dox 20 μM plus Cur 2 μM, Dox 20 μM plus Cur 4 μM, Dox 20 μM plus Cur 20 μM, Dox 20 μM plus Cur 40 μM,) were prepared via serial dilution with pipette off-chip. Pluronic F127 and Cell Tracker™ Green CMFDA Dye and EthD-1 were then added to the dissociated primary tumor cell suspensions of each mouse ($1.5 \times 10^6$ cells/ml) in tubes at a final concentration of 0.01% and 2 μM, respectively. The DMF chip was then filled with silicone oil (1 Cst). Cell suspensions and the drugs (Dox, Cur, Dox plus Cur) at a series of concentrations were loaded on-chip through inlet holes, moved under an actuation voltage to the neighboring electrodes sequentially toward the target cell culture electrodes, and mixed on-chip. The chips were then placed in an incubator for 24 h. Cell Tracker™ Green CMFDA Dye entered live cells and emitted green fluorescence. EthD-1 entered dead cells and emitted red fluorescence. Red fluorescent and Blue fluorescent images were captured under ×10 magnification using inverted fluorescent microscopy (Olympus). The absolute cell viability was calculated as the number of green cells/total number of both green and red cells. The relative cell viability under various drug treatments was normalized to the cell viability without the addition of drugs.

### MDA-MB-231 breast cancer xenograft mouse therapy

For single drug screening and therapy, according to the drug toxicity test results on the chip for each mouse, we divided the mice into three groups, with seven mice in each group, so a total of 21 mice were investigated. One group was injected with a relatively more effective drug (Cis), one with a relatively less effective drug (Wzb), and one injected with PBS as a no-treatment control. The injection mode was

intraperitoneal two times per week. The injection dose of Cis was 10 mg/kg. To keep the injection dose equal among the three groups, the treatment dose for the other two groups was 10 mg/kg. Before each injection time, the mice were weighed, and tumor volume was measured and calculated. The treatment period was one month.

For combinational drug screening and therapy, according to the drug toxicity test results on the chip for each mouse, we divided the mice into four groups, with five mice in each group. Noted that 7 mice in each group is ideal. Unfortunately, we did not get so many mice with required tumor sizes. So, we put 5 mice in each group. A total of 20 mice were investigated in this experiment. One group was injected with a single effective drug (Dox), one with a combinational effective drug (Dox plus Cur), one with a negative drug (Cur) and one injected with PBS as a no-treatment control. The injection mode was intraperitoneal two times per week. The injection dose of Dox and Cur was 10 mg/kg. In the combinational drug treatment group, the dose of Dox and Cur was 10 mg/kg, individually. To keep the injection dose equal among all the groups, the treatment dose for PBS control was also 10 mg/kg. Before each injection time, the mice were weighed, and tumor volume was measured and calculated. The treatment period was one month.

### Drug screening for clinical specimen on-chip
Tumor tissues (1.5–6 cm$^3$) were obtained after surgical hepatectomy at the Third Affiliated Hospital of Sun Yat-Sen University. The protocol used in this study was approved by the University of Macau' Research Ethics Board (Protocol # BRSERE21-APP015-IME). All the patients involved in this work have signed the consent forms before surgery. Immediately after surgery, the tumor specimens were transferred to the lab in 50-ml centrifuge tubes with 15 ml of preservation solution (DMEM/F12 plus 1% penicillin/streptomycin, 1% Glutamax, and 10 mM HEPES). The specimens were then taken out and minced into pieces of 0.1 cm$^3$ in a 10-cm culture dish in a biological safety cabinet. The minced tissues were incubated at 37 °C with EBSS (supplemented with 125 U/ml collagenase II and 0.1 mg/ml DNase I) in a culture dish for 2–6 h. The extent of digestion was observed, and the mixture was agitated using a pipette every half an hour to accelerate the digestion process until no large pieces of residue remained. Then, we added cold DMEM/F12 to stop the digestion and filtered the mixture through a 70-μm nylon cell strainer (Solarbio). The mixture was transferred into a 50-ml tube after filtration and spun for 5 min at 300 × $g$. Afterward, we resuspended the cells in red blood cell lysis buffer (Invitrogen™) for 2 min, added enough EBSS to stop lysis, and spun the tube. The cells were then stained with Cell Tracker™ Green CMFDA Dye (Invitrogen™). Subsequently, the cells (8 × 10$^5$ cells/ml) were resuspended in the medium containing 0.01% Pluronic F127 to promote smooth movement of the cell droplets on-chip. A series of concentrations (0 μM -DMSO control, 0.1 μM, 0.5 μM, 1 μM, 5 μM, 10 μM) of different drugs (Sor, Reg, Apa, and Len) were prepared and individually mixed with EthD-1. The final concentration of EthD-1 was 2 μM. We then loaded the cell droplets/drug droplets into the inlet holes and transferred the mixed drops to specific culturing electrodes for drug efficacy testing in an incubator for 24 h. The drug efficiency test was repeated thrice for each drug. The culture medium consisted of DMEM/F12, 1% penicillin/streptomycin, 1% Glutamax, 10 mM HEPES, 1:50 B2, 1.25 mM $N$-acetyl-l-cysteine, 10 mM nicotinamide, 50 ng/ml recombinant human EGF, 100 ng/ml recombinant human FGF10, 25 ng/ml recombinant human HGF, 10 μM forskolin, 5 μM A8301, 10 μM Y27632, and 3 nM dexamethasone. We observed and recorded fluorescent cells under a fluorescence microscope (Olympus) and calculated cell viability after treatment with different drugs. Cell Tracker™ Green CMFDA Dye entered live cells and emitted green fluorescence. EthD-1 entered dead cells and emitted red fluorescence. The absolute cell viability was calculated as the number of green cells/total number of both green and red cells. The relative cell viability under various drug treatments was normalized to the cell viability without the addition of drugs.

### Whole-exome sequencing
Genomic DNA was extracted from the tissues embedded in wax blocks. Exome sequences were captured according to BGISEQ-500 whole-exome enrichment and sequencing protocols. DNA from patients #1, #3, and #5 was subjected to whole-exome sequencing to screen and identify the variants at BGI Tech Solutions (Shen Zhen, China) using a BGISEQ-500 sequencer. The original sequence file for patients #1, #3, and #5 was shown in supplementary data 1, 2 and 3.

DNA was randomly fragmented to sizes mainly between 150 and 250 bp, end-repaired, ligated with adapters, and amplified with PCR for several cycles to infer the DNA libraries. A qualified capture library was obtained using the SureSelect Human All Exon kit (Agilent) and circularized. Rolling circle amplification was performed to produce DNA nanoballs, which were then loaded onto the BGISEQ-500 sequencing platform and high-throughput sequenced. Reads were aligned to the reference human genome sequence hg19 (GRCh37) using the Burrows-Wheeler Aligner, and HaplotypeCaller of GATK (v3.7) was applied for variant calling.

### Reporting summary
Further information on research design is available in the Nature Portfolio Reporting Summary linked to this article.

## Data availability
The authors declare that all datas supporting the findings of this study are available within the article, its Supplementary Information Files and from the corresponding author (Yanwei Jia: yanweijia@um.edu.mo) upon request. Source data are provided with this paper.

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

## Acknowledgements

This work is supported by the Macau Science and Technology Development Fund (FDCT) [FDCT 0029/2021/A1, Y.J.; SKL-AMSV(UM)-2023-2025, R.M.]; University of Macau [MYRG2020-00078-IME, Y.J.; MYRG-GRG2023-00034-IME, Y.J.]; Dr Stanley Ho Medical Development Foundation [SHMDF-OIRFS/2024/001]; Zhuhai Huafa Group [HF-006-2021]; the Key Program of National Natural Science Foundation of China (No. 62034005, H.Y.) and the National Natural Science Foundation of China (No. 61974084, H.Y.). We also thank the technical and administrative team of the State-Key Laboratory of Analog and Mixed-Signal VLSI at the University of Macau for all their support.

## Author contributions

Y.J., J.Z., and S.Y. conceived this project. Y.J., H.Y., P.M., and R.M. funded this project. J.Z. performed the experiments on a mouse model. Y.Liu. performed the experiments on clinical samples. W.J. designed the smart electrode share logic. Y.Li. H.L. X.Z., L.W., and N.Y. built the electrode controls. A.W. and P.C. helped in experiments in animal facilities. X.H., P.W., C.C., H.C., and S.Y. assisted in clinical sample preparation and data collection. J.Z., Y.L., and W.J. prepared the manuscript. Y.J., S.Y., H.Y., M.P., C.D., M.Y., T.H., and R. M. reviewed and edited the manuscript.

## Competing interests

The authors declare no competing interests.
