## [Peer Review File · Nature Communications]

Drug Screening on Digital Microfluidics for Cancer Precision MedicineREVIEWER COMMENTS

Reviewer #2 (Remarks to the Author):

The authors of this work developed a digital microfluidics chip for conducting drug screens on primary tumor biopsies. The results and contributions of the study can be categorized into two main areas: device design and biological improvements.

In terms of device design, the authors introduced a control board featuring an array of push-buttons that corresponded to the on-chip electrode array. This allowed for user-controlled droplet actuation. The control board also included dials and a display for managing and viewing actuation voltages. To enable what they called a "smart electrode-sharing protocol," the chip design incorporated a repeatable electrode pattern, allowing 24 inputs to control 96 electrodes. Additionally, the authors presented their DMF control circuit and high voltage generation system, consisting of push-buttons, mechanical relays, a signal generator, and a transformer.

The device presented in this work lacks novelty in several aspects. The analog push-button array, although a unique approach for creating a user interface for a DMF platform, does not significantly differ from previous proposals. It deviates from the automation capabilities of most other DMF devices, requiring manual actuation by users. This represents a step backwards compared to the traditional use of digital control. The electrical system detailed in Figure 2B is also not substantially different from previous designs (Alistar & Gaudenz, 2017; Fobel et al., 2013). The "smart electrode sharing protocol" described in the study is commonly known as "bussing" in the field of DMF, with modular bussing strategies and more complex modules having been demonstrated in prior research (Yang et al., 2008) (Perry et al., 2021). Furthermore, other solutions for improving droplet controllability on chip, such as using PCBs or active matrix control, have been previously explored (Anderson et al., 2021; Xing et al., 2021). While there is nothing inherently wrong with using established designs and strategies, the work should be appropriately placed in the Materials and Methods section or supplementary information, rather than being presented as results in the paper, as indicated by Figure 2 and 3, as well as lines 119-148 and lines 170-196.

Regarding biological improvements, the authors demonstrated a drug screening protocol for tumor cells obtained from biopsies. They initially conducted proof-of-concept screening using xenograft mice and successfully predicted the most effective drug for the mice. Subsequently, the authors employed their device to screen primary hepatocellular carcinoma specimens from five human patients. The on-chip screen identified a potentially successful drug for one patient out of the five, while no effective drugs were found for the remaining four patients. For three of the patients, whole exon sequencing was performed, confirming the absence of mutations in the genes corresponding to the drug targets tested on the chip, thereby validating the accuracy of the screen.

The drugs administered to the patients were done in a double-blind manner, meaning that neither the treating doctors nor the authors of the study were aware of the on-chip screen results until after the study. Two patients received drugs that were not screened on the chip, two patients did not receive any drug treatment, and one patient was treated with the drug recommended as most effective by the on-chip screen. After six months of follow-up, none of the patients experienced recurrence. The authors noted that the patient treated with the on-chip recommended drug had a higher likelihood of recurrence due to the metastatic nature of their cancer.

The biological novelty of the presented work is limited. The authors demonstrate previously shown biological assays, with the main claim of novelty being that their device requires fewer cells compared to other devices. However, the authors do not provide any quantification or data to support this claim. In the discussion of other microfluidic techniques for on-chip drug screening, the authors state that these systems inevitably waste precious biopsy samples, but they do not provide citations or experimental evidence to support this claim. Furthermore, the authors do not discuss the actual number of cells used or required in the cited works. The results of the study are notably lacking, with the most convincing results shown in Figure 4D and E.

However, it is worth noting that all human patients in the study, including those in the control and positive groups, were cancer-free after six months.

For the work to be suitable for publication in Nature Communications, the authors need to

address several points. Firstly, they should quantify the number of cells obtainable from biopsies at different stages of cancer progression to demonstrate the limitation of personalized drug screening. Secondly, they should quantify the number of cells required for assays in a 96-well plate and justify why these protocols cannot be adapted for a 384-well plate. Thirdly, they should quantify the number of cells required for channel-based microfluidic assays. Lastly, they should improve their results to establish a more conclusive relationship between device recommendations and clinical outcomes. This could be achieved by screening a broader range of drugs or combinations of drugs beyond what is possible with only well plates.

Specific Critiques

Line 59

Please specify how many cells (or what range of cells) can be obtained from a biopsy. It could be useful to know this information at numerous time points throughout the progression of a given cancer

Line 79

This is conjecture. I agree with the premise that syringes and tubing have dead volume and could lead to sample loss but without quantification (ie how many cells were required by the Eduati et al.) it can't be the sole justification for using DMF over droplet in channel

line 95-96

specify what is meant by "large amounts of primary cells" and "tiny biopsied primary tumors". Please use specific numbers or ranges

line 122-124

Why is it helpful to display real time actuation signal. I can't see why this would be of use to the end-user.

Line 141

Why would tailored droplet movements be useful in this setting?

Line 234-236

Please quantify what is a “limited number of primary tumor cells”. Please contrast this number with what is traditionally used and provide citations.

line 245-246

is it possible to get more than 10,000 cells from a single biopsy? Or was this the absolute limit? The entire novelty of this device hinges on the challenging nature of getting cells from a biopsy. It could be useful to provide information on exactly what was obtained from the mouse models as well as the human samples.

line 248

does this imply that ~3,300 cells were used per screen. If so - is that a device limitation? Can you use fewer cells to get an accurate screen? What is the least amount of cells you need for accurate analysis?

Line 321-324

What are the requirements to avoid hypoxia? How long can primary cells be exposed to hypoxic conditions before damage sets in.

Line 412-414

This statement needs a citation.

Lines 418-419

The authors claim their device’s “smart electrode sharing scheme” increases throughput but in comparison to what? A 96 well plate can handle far more unique reactions. Channel based microfluidics are significantly higher throughput. Even for a DMF chip this isn’t particularly high throughput.

Lines 420-421

What do the authors mean by “automatic and intelligent”

The analog push buttons seem to remove the possibility of automation from this system

Lines 421-423

The authors present estimates for how long it takes to complete an experiment and an estimated cost per sample but without comparisons to well-plate assays these estimates are without context.

References:

Alistar, M., & Gaudenz, U. (2017). Opendrop: An integrated do-it-yourself platform for personal use of biochips. *Bioengineering*, 4(2).

<https://doi.org/10.3390/bioengineering4020045>

Anderson, S., Hadwen, B., & Brown, C. (2021). Thin-film-transistor digital microfluidics for high value in vitro diagnostics at the point of need. *Lab on a Chip*, 21(5), 962–975.

<https://doi.org/10.1039/d0lc01143f>

Fobel, R., Fobel, C., & Wheeler, A. R. (2013). DropBot: An open-source digital microfluidic control system with precise control of electrostatic driving force and instantaneous drop velocity measurement. *Applied Physics Letters*, 102(19). <https://doi.org/10.1063/1.4807118>

Perry, J. M., Soffer, G., Jain, R., & Shih, S. C. C. (2021). Expanding the limits towards 'one-pot' DNA assembly and transformation on a rapid-prototype microfluidic device. *Lab on a Chip*.

<https://doi.org/10.1039/d1lc00415h>

Xing, Y., Liu, Y., Chen, R., Li, Y., Zhang, C., Jiang, Y., Lu, Y., Lin, B., Chen, P., Tian, R., Liu, X., & Cheng, X. (2021). A robust and scalable active-matrix driven digital microfluidic platform based on printed-circuit board technology. *Lab on a Chip*, 21(10), 1886–1896.

<https://doi.org/10.1039/d1lc00101a>

Yang, H., Fan, S.-K., & Hsu, W. (2008). Connecting interface for modularization of digital microfluidics. *Microfluidics, BioMEMS, and Medical Microsystems VI*, 6886(February 2008), 68860L. <https://doi.org/10.1117/12.765652>

Reviewer #3 (Remarks to the Author):

Zhai et al have developed a DMF device for drug testing on cancer cells.

General comments:

1. The whole manuscript needs to be thoroughly checked for grammar and typo errors to increase the readability and to avoid minor errors. Few examples though not all are given

below:

a. P19: “fellow up”

b. many more.

2. In this study, mostly the effect of single best drug is studied with a single cell type.

However, cancers have multiple cell types and most of them require a combinatory treatment of a number of drugs. As many reports show a combinatory drug approach, the current study is very basic to assume one drug-one cell approach of cancer treatment.

Kindly refer the literature¹ for a detailed study and how the current literature provides better data.

3. The primary cells need to be characterized something similar to the published literature² for better understanding of readers.

Specific comments:

1. P4 line96: it was mentioned in introduction section that “the biopsied primary tumors has never been tested for DMF”. It appears that this is the aim of the current study.

However, the biopsied samples are not tested rather some cells isolated in 2D are tested, which was not in line with recent knowledge about cancer-drug testing.

2. P4 Line 102: the term “xenograft mouse breast cancer” is very confusing and the same meant term is used in different forms in the article. It appears as mouse breast cancer cells, which the readers might confuse to find in actual literature. Kindly use same and uniform term throughout the manuscript.

3. Figure 1: the dissociated liver tissue and other xenograft tissue provides cancer cells. However, it is fully unclear the properties of such cells and whether they are cancer stem cells and what are the surface markers and growth potential. It needs to be fully characterized to understand the scientific merits of this study.

4. P6 line 155: how the cell culture spots are pre-treated with gelatin without the other parts is not clear.

5. Video s3 and s4: is fully not comprehensible. The authors are requested to provide live fluorescent microscopic pictures to show the cells are moving as the cells are marked with a fluorescence dye.

6. P11 line 246: Each biopsy sample is thought to contain 10000 cells. However, in supplementary data, it is mentioned that each area was injected 20000 cells (MDA-MB-231 cell suspensions (2×10^6 cells/ml) (100 μ l)) to start with. As these data are not clear, kindly

provide the quantitative data of how many cells are started and what is the end stage of number of cells in a fully grown cancer tissue.

7. Table 1: the + sign shows cell viability of 80% as written in caption. This is arbitrary and a quantitative image should be provided to each of the data.

8. P15 line 333: the sentence “since the tumor is resected, the tumor recurrence is used as a marker” is a controversial sentence. Usually the tumor is resected only when there is no chance of recurrence and only a small percentage of tumors show recurrence. otherwise the tumor is considered not suitable for surgery. Kindly explain.

9. In the above, the histological pictures are required to show the tumor resection and the behavior of the original cancer after resection that it is clearly resected.

10. Table 2: what is exactly mutated in which exons. The original data needs to be deposited with analysis for readers to understand.

11. P18 line 382: the angiogenesis affected genes should be quantified and shown here. Again the previous study should give a guideline as mentioned in general remarks.

12. P19 line 439: as currently the 3D culture data is foremost, a general discussion section should be devoted how this should be applicable to 3D cancer organoids. It is one of the most important limitations of the current study.

13. Supplementary line 80: “tumors larger or smaller are discarded”. This is very ambiguous and not in line with scientific publications. Kindly mention all the data discarded.

14. For all data, a corresponding fluorescent images are required for readers to understand. IN this manuscript, only quantitative data is given without any real images of cells being stained in all the scenarios.

Literature 1: <https://www.nature.com/articles/s41467-021-25921-9>

Literature 2: <https://pubs.rsc.org/en/content/articlehtml/2023/bm/d2bm01870e>

RESPONSE LETTER

RESPONSES TO REVIEWER #2

Thanks for the reviewer's valuable suggestions that have helped us to improve the manuscript significantly. Following the reviewer's suggestions, we have run more experiments on combinational drug screening in the DMF system, quantified the numbers of cells needed for the on-chip drug screening, discussed more literature for comparison, and reorganized the manuscript for a better presentation. The revised parts are highlighted in blue in the manuscript and SI. Detailed responses to each comment are shown below.

Reviewer #2 (Remarks to the Author):

Comment 1:

The authors of this work developed a digital microfluidics chip for conducting drug screens on primary tumor biopsies. The results and contributions of the study can be categorized into two main areas: device design and biological improvements.

In terms of device design, the authors introduced a control board featuring an array of push-buttons that corresponded to the on-chip electrode array. This allowed for user-controlled droplet actuation. The control board also included dials and a display for managing and viewing actuation voltages. To enable what they called a "smart electrode-sharing protocol," the chip design incorporated a repeatable electrode pattern, allowing 24 inputs to control 96 electrodes. Additionally, the authors presented their DMF control circuit and high voltage generation system, consisting of push-buttons, mechanical relays, a signal generator, and a transformer.

The device presented in this work lacks novelty in several aspects. The analog push-button array, although a unique approach for creating a user interface for a DMF platform, does not significantly differ from previous proposals. It deviates from the automation capabilities of most other DMF devices, requiring manual actuation by users. This represents a step backwards compared to the traditional use of digital control. The electrical system detailed in Figure 2B is also not substantially different from previous designs (Alistar & Gaudenz, 2017; Fobel et al., 2013). The "smart electrode sharing protocol" described in the study is commonly known as "bussing" in the field of DMF, with modular bussing strategies and more complex modules having been demonstrated in prior research (Yang et al., 2008) (Perry et al., 2021). Furthermore, other solutions for improving droplet controllability on chip, such as using PCBs or active matrix control, have been previously explored (Anderson et al., 2021; Xing et al., 2021). While there is nothing inherently wrong with using established designs and strategies, the work should be appropriately placed in the Materials and Methods section or supplementary information, rather than being presented as results in the paper, as indicated by Figure 2 and 3, as well as lines 119-148 and lines 170-196.

Response 1:

Thank you for your precise summary and valuable suggestions for reorganizing the manuscript.

About the automatic control, yes, quite a few works have been published for automatic control of droplet movement, including our own works¹⁻⁶. We agree with you that the push-button control is a step backwards for involving human interference during the whole process. The reason we took a step backward was to fit the complicated situation in handling the primary tumor cells drug screening. In the automatic controls, either a set of electrode charging steps have been programmed for completing a series of steps that lack error-tolerance, or another screening signal was provided in addition to the actuation signal for real-time identification of droplet location which adds another level of electronic signal. However, primary tumor cells are more vulnerable than commercialized cancer cells. Especially when screening various drugs with different viscosity, hydrophobicity, or surface tension values, we noticed that programmed automatic control frequently failed to complete the task. On the other hand, we wanted to minimize the influence of electric signals on the primary tumor cells, which also excluded the real-time droplet location identification. The push-down button design allowed us to monitor where the drop was and adjust the actuation to make sure all the droplets went to the location they should go, and therefore did not waste any clinical tumor cells. Furthermore, the system is small enough that it can be taken into the animal facility and surgical operation room in the hospital for the most fresh tumor cells. At the same time, we are developing a more reliable automatic control system including an on-chip drug delivery for drug screening. We hope that it will make the system small, the control safe, and the operation robust for non-specialists in the future. In this paper, we focus on validating that drug screening in the DMF system of primary tumor cells can be a reliable instruction for precision medicine. We have added these discussions in the manuscript on page 15, 16, line 460-475, and references 54-60 for clarification.

Regarding the electrical system, we agree with you that the design is not substantially different from previous designs^{3,7}. We have moved the detailed description of the system design into the SI. A brief introduction of the system's electrical design is added in SI on page 2, lines 35-54 for reference to the readers.

Regarding the "smart electrode sharing protocol", we compared our work with the work of Yang et al. and Perry et al. In Yang's work⁸, the structure they proposed can only be used to control the two-way movement of the droplets. In order to make the DMF chip more suitable for high-throughput drug screening, we extended it perpendicular to the bus. The extended repeating unit not only guarantees the fixed number of control electrodes, but also ensures the minimization of the total number of electrodes required to achieve the miniaturization and portability of the entire drug screening system. Based on the chip structure we proposed, we further designed a templated control logic scheme for the valuable drug screening task, which can determine the control logic of drug screening in a constant time. The proposed method avoids the overhead using complex algorithms to calculate the control logic, and can ensure the correctness of the drug screening function. In Perry's work⁹, they investigated repetitive electrode units but not signal sharing between electrodes. In our work, the repeatable units also include a unique wiring structure that ensures that

they are connected to the same control pin. Thus, the number of control pins is fixed even when the number of repeat units increases. However, in Perry's work, the number of control pins increases proportionally when the number of repeating units increases. We have added the discussion of their work and the difference on pages 6, line 154-169. The Smart electrode-sharing protocol for drug screening has been added in SI on page 3, 4, line 73-107.

Regarding using other substrates such as PCB for convenient and powerful controllability like active matrix control, we actually found glass substrate the most appropriate material. For example, the surface flatness of PCB does not meet the requirement of single-cell observations. As shown in Fig. R1, the scratch on the copper electrodes would cover the signal of cells, making it difficult for cell identification. LCD could be another option with active matrix control, but the cost has limited its wide application for biological samples. So, we chose a glass substrate and combined it with the electrode-sharing algorithm to achieve the same control ability with fewer control signals. We have added the discussion in SI on page 4, 5, lines 113-120.

Fig. R1 The image result of PCB substrate observed under natural light (a) and the enlarged substrate area observed under microscopy (b).

Comment 2:

Regarding biological improvements, the authors demonstrated a drug screening protocol for tumor cells obtained from biopsies. They initially conducted proof-of-concept screening using xenograft mice and successfully predicted the most effective drug for the mice. Subsequently, the authors employed their device to screen primary hepatocellular carcinoma specimens from five human patients. The on-chip screen identified a potentially successful drug for one patient out of the five, while no effective drugs were found for the remaining four patients. For three of the patients, whole exon sequencing was performed, confirming the absence of mutations in the genes corresponding to the drug targets tested on the chip, thereby validating the accuracy of the screen.

The drugs administered to the patients were done in a double-blind manner, meaning that neither the treating doctors nor the authors of the study were aware of the on-chip screen results until after the study. Two patients received drugs that were not screened on the chip, two patients did not receive any drug treatment, and one patient was treated with the drug recommended as most effective by the on-chip screen. After six

months of follow-up, none of the patients experienced recurrence. The authors noted that the patient treated with the on-chip recommended drug had a higher likelihood of recurrence due to the metastatic nature of their cancer.

The biological novelty of the presented work is limited. The authors demonstrate previously shown biological assays, with the main claim of novelty being that their device requires fewer cells compared to other devices. However, the authors do not provide any quantification or data to support this claim.

In the discussion of other microfluidic techniques for on-chip drug screening, the authors state that these systems inevitably waste precious biopsy samples, but they do not provide citations or experimental evidence to support this claim. Furthermore, the authors do not discuss the actual number of cells used or required in the cited works. The results of the study are notably lacking, with the most convincing results shown in Figure 4D and E.

Response 2:

Thanks for your precise summarization of our work and pointing out the issues in our presentation.

We have reviewed more literature and run more experiments to quantify the cell numbers needed for drug screening on DMF chips. The total cell numbers from a #16 needle (11 mm sample groove) is about 1.5×10^4 for mice samples and 5×10^4 for #18 needle (18 mm sample groove), as shown in Fig. R2a and Fig. R2b. However, as reported in Repetto's work¹⁰, the cells needed for drug screening on 96-well plate are about 2.5×10^4 to 5×10^5 cells per condition. For a screening with 6 drug concentrations, 1.5×10^5 to 3×10^6 cells would be needed. Sirenko et al tried drug screening on 384-well plate¹¹, they used totally 7×10^4 cells for a 6 condition screening. The number of primary tumor cells obtained with a biopsy needle is far less than enough for drug screening in traditional microplates.

Several works have reported microfluidics for drug screening^{12, 13}. In Wong's work¹², they reported that 1.5×10^6 cells were required for 96-well plate, 4.0×10^5 cells for 384-well plate, and 16,000 cells for the PDMS chip under the per drug per dose condition. In Eduati's work¹³, they reported that about 100 live cells were required for each condition. As can be seen from the data, microfluidics can significantly lower the required cell numbers for drug screening. So far, the drug screening was all run on channel-based microfluidics, which requires bulky and complicated controls including external pumps and connection tubes etc.¹²⁻¹⁴. In Eduati's work, they reported that about 100 live cells were required in a droplet, without mentioning how much percentage of the input cells were captured in droplets with drugs. In our experience with channel microfluidics^{15, 16}, the droplets were nonuniform for the first few minutes due to unstable flow rates and pressures. Those droplets were normally discarded for analysis. So we anticipated the same thing would happen in their setup. In addition, in order to solve the problem of the swept volume introduced by syringes and tubing, Werner developed a novel microfluidic droplet generator system, which consisted of a series of peristaltic pumps controlled by an integrated pneumatic logic circuit for reagents to be consumed

directly from a well plate¹⁷. The fabrication of the system was tedious and complicated. The initial nonuniform droplet generation and connection tube would inevitably waste some biopsy samples, making available cells even less than obtained. DMF system can manipulate a single droplet with precise control to use up all the biopsied cells.

On DMF chip, we usually used about 300 cells in each droplet for drug screening, but this may not be the lowest number that can provide reliable drug screening results. To test the limit of cell numbers for valid drug screening, we ran a serial dilution of cell numbers from 100 to 100,000 in the presence of the drug in a 96-well microplate or on-chip. MDA-MB-231 was used as the cell model, and EP as the drug model. As shown in Fig. R2c, the cell viability decreased with increasing the drug concentration in each group of different cell numbers. The IC₅₀ values were comparable when the cell number was more than 1000. When the cell number was lowered to 500 cells per sample, the deviation was obviously enlarged, and the IC₅₀ value increased a lot from 9 μM to 29 μM. The reliable cell required for 96-well plate is about 1000 cells.

We further tested the drug screening with 50, 100, 200, 300 cells in a droplet on a DMF chip. No higher cell numbers were tested due to the droplet accommodation. As shown in Fig R2d, the cell viability curve was similar from 100 cells to 300 cells. We expect the DMF chip can handle as least as 100 cells per condition. This number is consistent with that reported by Eduati et al. for microfluidic drug screening¹³.

We have added the discussion on page 3 (lines 81-89), page 16 (lines 476-484) in the manuscript, and pages 13, 14, lines 329-364 in the SI.

Fig R2. Cell count of each biopsies from mice (a) and human patients (b). (c) Cell Viability results of a series of MDA-MB-231 cell numbers (1.0×10^5 cells, 1.0×10^4 cells, 5.0×10^3 cells, 10^3 cells, 5.0×10^2 cells) after drug EP (0 μM , 5 μM , 10 μM , 25 μM , 50 μM) treatment based on well-plates method. (d) Cell Viability results of a series of MDA-MB-231 cell numbers (300 cells, 200 cells, 100 cells, 50 cells) after drug EP (0 μM , 10 μM , 20 μM , 30 μM , 40 μM , 50 μM) treatment based on chip method.

Comment 3:

However, it is worth noting that all human patients in the study, including those in the control and positive groups, were cancer-free after six months.

Response 3:

Yes, all the five patients were cancer-free after six months. That mainly comes from the criteria of surgery and medication in hospitals. In clinical experiments, we cannot treat patients as mice for ethical reasons, giving negative drugs to a patient when we know it most probably would have no effect. Therefore, the experiments were conducted double-blinded. The doctor treated the patients based on the clinical blood test, imaging results, tumor conditions, etc. for the best prognosis. The two cases that fell in the control group were not treated with any drug after surgery because the doctor predicted no recurrence of tumor in the near future. The patient who happened to receive the effective drug had the worst conditions (metastatic tumor, immune suppression, old age) and was prone to tumor recurrence. With these conditions, cancer recurrence occurs typically within 6 months. The drug description was a try with good hope. Cancer-free in this positive

group patient means more than cancer-free in the other two patients.

We also agree with the reviewer that the cancer-free in the control group made the results less convincing. So, we ran the sequencing of primary tumor samples to test whether they have specific genetic mutations related to those drugs. The sequencing results were consistent with the drug screening results, validating the on-chip drug screening results in another way.

For clinical precision medicine, we expect to run more samples for a more convincing result. We have added the discussion of the limitations of the current system on page 13 for a better understanding of the technique.

Comment 4:

For the work to be suitable for publication in Nature Communications, the authors need to address several points. Firstly, they should quantify the number of cells obtainable from biopsies at different stages of cancer progression to demonstrate the limitation of personalized drug screening. Secondly, they should quantify the number of cells required for assays in a 96-well plate and justify why these protocols cannot be adapted for a 384-well plate. Thirdly, they should quantify the number of cells required for channel-based microfluidic assays. Lastly, they should improve their results to establish a more conclusive relationship between device recommendations and clinical outcomes. This could be achieved by screening a broader range of drugs or combinations of drugs beyond what is possible with only well plates.

Response 4:

We greatly appreciated the reviewer's valuable suggestions.

Firstly, we have quantified the number of cells obtained from biopsies at different stages of cancer progression. Two types of biopsy needles were used in this work. Needle #16 (11 mm sample groove) was used in mice samples, and needle #18 (18 mm sample groove) was used in clinical samples. As shown in Fig. R2a, about 1.5×10^4 cells were obtained from the xenograft tumor on mice. To quantify the cell numbers obtained from different stages of cancer progress, we obtained samples from three liver cancer patients with two at early stage and one at late stage. As shown in Fig. R2b, the cell numbers do not correlate with the stage of tumors, ranging from 3.8×10^4 cells to 6.6×10^4 cells, with an average cell number of 5×10^4 . This is reasonable because the obtained cell number depends on the groove volume on the biopsy needle, not the sample stage. We have added the data and discussion on page 6, 7, lines 180-186, and Fig. S5 in the SI.

Secondly, the minimum required cell number for a reliable drug screening in a 96-well plate was quantified to be 1000, as shown in Fig. R2c. Compared to the 100 μL solution volume in 96-well plate, 384-well plate takes 50 μL solutions per sample. The biopsied cells may be enough for a screening of one drug with 6 conditions in a 384-well plate. One drug screening would not provide a useful information for precision medicine. We have added the discussion on page 13, 14, lines 329-364 in the SI.

Thirdly, we do not have facilities to carry out channel-based microfluidic drug screening assays. From

literature, in one work¹², the author reported that 16,000 cells for PDMS chip for each condition. In another work¹³, the author reported that about 100 live cells were used for each condition. Given the channel microfluidics has similar droplet size, we expected the required cell numbers would be similar to those on DMF chip (at least 100 cells). Our data also confirmed that 100 cells can give a similar result as 300 cells, as shown in Fig. R2d. We have added the discussion in SI, on page 14.

Lastly, we tried to build a more conclusive relationship between the chip recommendation and precision medicine results. Since the clinical samples were difficult to obtain and took longer time for precision medicine outcomes, we screened more drugs and combinations of drugs in mouse models. In the previous experiments, Cis and Wzb were screened in a single drug mode. In the supplement experiments,

Fig. R3 On-chip single drug and combinational drug screening results of biopsy samples from 15 individual mice. The drugs are Dox, Curcumol and Dox plus Curcumol.

Fig. R4. (a) On-chip drug screening results for the biopsy samples from 15 mice with Dox (10 μ M) or Cur (20 μ M) or Dox (10 μ M) plus Cur (20 μ M) treatment and the corresponding drug administration mode and therapeutic effect *in vivo*. (b) Representative on-chip drug screening fluorescent imaging results for the biopsy samples from mouse 4, 9 and 13 after Dox (10 μ M) or Cur (20 μ M) or Dox (10 μ M) plus Cur (20 μ M) treatment (c, d) The results of mice treatment. The relationship between drug administration times and (C) tumor volume, (D) mouse body weight.

Doxorubicin (Dox) and Curcumol (Cur) were used as the drug models. The biopsy samples from 15 mice were used for on-chip drug screening. Guided by the on-chip screening results, the mice were sorted into three groups, with one group treated with a single effective drug, one group with a single least effective drug, and the other group with the combinational drug. Another group of 5 mice was treated with PBS solution as a control group.

Fig. R3 shows the on-chip drug screening results of Dox alone, Cur alone, and the combination of 10 μ M Dox and various concentrations of Cur. As can be seen, in most of the cases, the drug combination with an additional 10 μ M Dox killed more cancer cells than Cur alone.

Fig. R4a summarizes the cell viability at different concentrations of Dox, Cur, or Dox & Cur. As can be seen from Fig. R4a, Dox alone worked better than Cur alone in most cases with lower cell viability. The combination of two drugs always showed better effects than a single drug, with lower cell viability. Fig. R4b shows representative on-chip drug screening fluorescent images of the biopsy samples from mouse 4, mouse 9, and mouse 13.

To test the correspondence between on-chip drug screening and *in-vivo* drug therapy, one group of mice was treated with 10 mg/kg Dox as the positive drug group, one group of mice was treated with 10 mg/kg Cur as the negative drug group, and one group was treated with the combination of 10 mg/kg Dox and 10 mg/kg Cur as the combination drugs group.

Fig. R4c shows that the tumor volumes of the positive and combinational drugs groups were smaller than those of the negative drug and control groups after one month of treatment, aligning with the on-chip drug screening results. The tumor growth trend within the treatment period differed among the mice within the same group, suggesting the existence of individual differences. In the positive drug group, the tumor volume of combinational drug Dox & Cur treated mice was smaller than those treated with single drug Dox, which was consistent with the on-chip drug screening result and suggested the higher therapeutic effect for the combinational treatment.

Body weight measurement was further used to evaluate the drug toxicity. As shown in Fig. R4D, no considerable changes in body weight were observed in the mice of the negative drug and control groups. However, in the positive drug group, the body weight of the mice reduced to some extent, suggesting the potential toxicity of the effective drug. These results collectively demonstrate the reliability of the DMF platform for on-chip combinational drug screening usage.

We have added the combinational drug screening part on pages 9, 10 in the revised manuscript and supporting information pages 8, 9, and 16.

Specific Critiques

Line 59

Please specify how many cells (or what range of cells) can be obtained from a biopsy. It could be useful to know this information at numerous time points throughout the progression of a given cancer

--- Thanks for your suggestion. As can be seen from the response to comment 4, there are around 5×10^4 cells in a biopsy sample. We have put in the specific number in the revised manuscript. Please refer to page 2, line 60. The data has been added to SI, Fig. S5.

Line 79

This is conjecture. I agree with the premise that syringes and tubing have dead volume and could lead to sample loss but without quantification (ie how many cells were required by the Eduati et al.) it can't be the sole justification for using DMF over droplet in channel.

---We thank for the reviewer's comment. In Eduati's work, they reported that about 100 live cells were required in a droplet, without mentioning how much percentage of the input cells were captured in droplets with drugs. In our experience with channel microfluidics^{15, 16}, the droplets were nonuniform for the first couple of minutes due to unstable flow rates and pressures. Those droplets were normally discarded for analysis. So we anticipated the same thing would happen in their setup. In addition, in order to solve the problem of the swept volume introduced by syringes and tubing, Werner developed a novel microfluidic droplet generator system, which consisted of a series of peristaltic pumps controlled by an integrated pneumatic logic circuit, for reagents to be consumed directly from a well plate¹⁷. However, the fabrication of the system is tedious and complicated. We agree with you it is a conjecture without quantified data. We have revised the sentence to emphasize the convenient operation of DMF without the pumps or valves as in channel microfluidics to validate the advantages of DMF. Please refer to page 3, lines 81-89, and references 16-18.

line 95-96

specify what is meant by "large amounts of primary cells" and "tiny biopsied primary tumors". Please use specific numbers or ranges

--- We thank for the reviewer's suggestion. The primary cells isolated from organs model system, such as porcine aortic endothelial cells, or porcine aortic valvular interstitial cells, can undergo multiple sub-cultures as a cell line¹⁸. Theoretically, any cell amounts can be achieved by passing the cells through generations to be used for drug screening on-chip or not. For biopsied primary tumors, the cell number was less than 10^5 . We have clarified the concept with specific numbers, as shown on page 4, lines 104-107.

line 122-124

Why is it helpful to display real time actuation signal. I can't see why this would be of use to the end-user.

--- This is only to provide additional information on how much voltage was applied to actuate the droplet

with a specific drug, so that similar voltage can be set directly for the same drug. It is not a mandatory request for operation. We have revised it for clarification on page 2, lines 40-42, in the SI.

Line 141

Why would tailored droplet movements be useful in this setting?

--- This is a great question! As mentioned in the response to comment 1, we actually used automatic control for droplet transportation when starting the project. However, we found out that a program that worked for one drug may fail for another. We suspected it was because of the diverse properties of different drugs. We made a step backward for push-button control, so that we can change the actuation voltage, actuation time, and the droplet movement path to ensure each drop can safely reach the position it should go. We have revised the description for clarification on page 3, lines 58-63, in the SI.

Line 234-236

Please quantify what is a “limited number of primary tumor cells”. Please contrast this number with what is traditionally used and provide citations.

--- The number of primary tumor cells obtained with a biopsy needle was from 1×10^4 to 5×10^4 as shown in Fig. R2a and Fig. R2b, depending on the gauge of the biopsy needle. Traditionally 96-well or 384-well plates used 1.5×10^6 cells or 4.0×10^5 cells respectively as reported in literature¹⁹. We have revised the manuscript for clarification. Please refer to page 6, lines 180-182 in the manuscript. References have been updated as well, Ref 8.

line 245-246

is it possible to get more than 10,000 cells from a single biopsy? Or was this the absolute limit? The entire novelty of this device hinges on the challenging nature of getting cells from a biopsy. It could be useful to provide information on exactly what was obtained from the mouse models (HE staining result) as well as the human samples.

--- Yes, we could get more than 10,000 cells from a single biopsy. The obtained primary tumor cell number is affected by the size of the biopsy needle and the size of the tumor. As reported, the obtained primary tumor cell number is within the range of 10^4 - 10^6 cells^{12, 19, 20}. In our work, the number of cells ranges from 1×10^4 to 5×10^4 with gauge #16 biopsy needle for the biopsies from mice and #18 biopsy needle for the biopsies from human patients, as shown in Fig. R2a and Fig. R2b.

To verify what was obtained from clinical liver cancer biopsies, we ran a flow cytometry analysis. In this experiment, the combination of CD44 and CD24 was used to identify the existence of cancer stem cells (CSC)²¹. Cells with high CD44 expression and low CD24 expression were considered as CSC. A high expression is indicated by the right shift of intensity peak of stained cells compared to unstained cells. As shown in Fig. R5a and Fig. R5b, the CD44 peaks of the stained cells from two biopsy samples were right-shifted from the unstained cells, which indicated a high expression of CD44 in both samples. In Fig. R5a

and Fig. R5b, the peak positions of both samples had no noticeable difference for stained and unstained cells, indicating a low expression of CD24. The high expression of CD44 and low expression of CD24 demonstrated that there were CSCs obtained in the biopsy samples.

We also ran an analysis of CD34, CD45, CD14, CD19, and HLA-DR, which are associated with Hematopoietic and immune cells²². As shown in Fig. R5a and Fig. R5b, the peaks were similar for all the markers and showed no difference for stained and unstained cells, indicating no much immune cells existed in the biopsy sample. Based on these data, we concluded that the drug screening of the biopsy sample mainly reflected the tumor cells' responses to the screened drugs.

To verify the obtained cells in the biopsy of mouse models, we ran HE staining for analysis. As shown in Fig. R5c, the similar morphology of cells in the slides, with a big nuclear to cytoplasmic ratio, suggested almost all of the cells were tumor cells. We have added the discussion in the manuscript and the data in the SI. Please refer to pages 13, 14, Fig. 7 in the manuscript, and Fig. S8 on page 18 in the SI.

Fig. R5. (a, b) Flow cytometry analysis of patients#6, #7, revealed cells with high CD44 but low CD24 expression. (c) HE staining result of the biopsy samples from two mice.

line 248

does this imply that ~3,300 cells were used per screen. If so - is that a device limitation? Can you use fewer cells to get an accurate screen? What is the least amount of cells you need for accurate analysis?

--- As shown in Fig. R2a and Fig. R2b, about 5×10^4 cells were obtained in a biopsy sample. During the process, some cells may be lost due to centrifugation or half-filled biopsy needles etc. So, roughly 500 to 1,000 cells per dose and 3,000 to 6,000 cells per drug were used for screening on-chip. This number is not the device limitation. What we needed was a cell density with reliable cell viability assessment. In Eduati's work¹³, they reported about 100 live cells for screening per dose per drug in a 500 nL droplet in channel microfluidics. We anticipated we could reach the similar amount of cells per droplet on DMF for reliable results. We could achieve this by adjusting the size of electrodes on the DMF chip so that the size of the droplet is comparable to the droplet in channel microfluidics.

We have added the discussion in the manuscript. Please refer to page 16, lines 476-494.

Line 321-324

What are the requirements to avoid hypoxia? How long can primary cells be exposed to hypoxic conditions before damage sets in.

--- To avoid hypoxia, the requirements are to maintain sufficient oxygen supply and eliminate factors that consume oxygen. Primary cells can be exposed to hypoxic conditions for a certain period before damage occurs, but this varies depending on cell type and experimental conditions. Generally speaking, the longer the exposure time, the greater the risk of damage to cells.

To figure out how long can primary tumor cells be exposed to hypoxic conditions before damage sets in, we designed an experiment. A relatively large tumor tissue with good initial activity was chosen and put in a 50 ml tube with no air in tube to create hypoxic conditions (Fig. R6a). Then we cut off a piece of tumor tissue at different time points (0h, 6h, 12h), dissociated them into single cells, and checked the cell activity. As shown in Fig. R6b, the cell viability remained at about 70% when the primary liver cancer cells were kept in a relatively hypoxic environment for 6 hours. There was no significant difference between 0 hours and 6 hours. However, cell viability decreased sharply after 6 hours and was less than 10% at 12 hours (Fig. R6c). This indicates that damage occurs after 6 hours of hypoxia condition.

For the highest cell viability in drug screening, the freshest cells have the best results. The most extended period for the tumor to be kept without any treatment was 6 hours. We have added the discussion on page 17 in the SI. The data has been put in the SI for reference. Please refer to Fig. S7.

Fig. R6. (a). Picture of tumor tissue from patient #9 in hypoxia condition. (b). Corresponding chart showed cell viability from primary liver cancer samples after exposed to hypoxia condition for 0-12 h. (c). Fluorescent image results of dissociated cells from primary liver cells after exposed to hypoxia condition for 0-12 h. Green represents living cells and red represents dead cells. Scale bars are 100 μm .

Line 412-414

This statement needs a citation.

--- We thank the reviewer's suggestion. Lee et al²³, Mathur et al²⁴, and Shi et al²⁵ have validated using cell drug screening for precision medicine. We have added the citations on page 14, line 422 and references 51-53.

Lines 418-419

The authors claim their device's "smart electrode sharing scheme" increases throughput but in comparison to what? A 96 well plate can handle far more unique reactions. Channel based microfluidics are significantly higher throughput. Even for a DMF chip this isn't particularly high throughput.

--- We thank for the reviewer's comment. The throughput was compared with a traditional DMF chip, which employs direct addressing with one control pin for one electrode control. With traditional control, 24 actuation signals can only operate 24 electrodes, while the sharing scheme in this work allows 24 actuation signals to control 96 electrodes. More drugs would be able to be screened on-chip with the same amount of control signals. We agree with you that the description was unclear. We have clarified that the smart electrode sharing scheme increases the drug screening throughput compared with direct addressing control. Please refer to page 14, lines 426-430.

Lines 420-421

What do the authors mean by “automatic and intelligent”

The analog push buttons seem to remove the possibility of automation from this system

--- We agree with you. This claim has been deleted in the revised manuscript.

Lines 421-423

The authors present estimates for how long it takes to complete an experiment and an estimated cost per sample but without comparisons to well-plate assays these estimates are without context.

--- Thanks for your comments. The amount of primary tumor cells is insufficient for traditional drug screening in microplate. Culturing of the primary cells would be needed to generate enough cells for drug screening, which may take more days besides the 24 hours drug screening. With our system, the primary tumor cells were directly screened on-chip. The entire workflow, including cell dissociation, drug screening, and data analysis, can be completed in 36 hours. More critically, the sub-culture of the primary cells may introduce unexpected mutations in descendant cells and affect the drug screening results, while the primary tumor cell drug screening has no such risk.

About the price, the net cost of the microplate (\$ 1.5 for a 96-well plate) is low compared with a DMF chip (\$20 per chip). However, drugs are more expensive. For example, Len costs \$140 and Ep costs \$160 for 10 mg. To maintain the same drug concentration in a 96-well plate with 100 μ L solution would cost 300 times more than that on a DMF chip with 0.3 μ L solution. Furthermore, the DMF chip can be washed and surface-treated again for reuse. Suppose one DMF chip can be reused 10 times, the cost per drug condition would be lowered by 3000 times comparable with microplate.

We have added the discussion on the comparison with the well-plate for clarification on page 15, lines 438-445.

References:

Alistar, M., & Gaudenz, U. (2017). Opendrop: An integrated do-it-yourself platform for personal use of biochips. *Bioengineering*, 4(2). <https://doi.org/10.3390/bioengineering4020045>

Anderson, S., Hadwen, B., & Brown, C. (2021). Thin-film-transistor digital microfluidics for high value in vitro diagnostics at the point of need. *Lab on a Chip*, 21(5), 962–975. <https://doi.org/10.1039/d0lc01143f>

Fobel, R., Fobel, C., & Wheeler, A. R. (2013). DropBot: An open-source digital microfluidic control system with precise control of electrostatic driving force and instantaneous drop velocity measurement. *Applied Physics Letters*, 102(19). <https://doi.org/10.1063/1.4807118>

Perry, J. M., Soffer, G., Jain, R., & Shih, S. C. C. (2021). Expanding the limits towards ‘one-pot’ DNA assembly and transformation on a rapid-prototype microfluidic device. *Lab on a Chip*. <https://doi.org/10.1039/d1lc00415h>

Xing, Y., Liu, Y., Chen, R., Li, Y., Zhang, C., Jiang, Y., Lu, Y., Lin, B., Chen, P., Tian, R., Liu, X., & Cheng, X. (2021). A robust and scalable active-matrix driven digital microfluidic platform based on printed-circuit board technology. *Lab on a Chip*, 21(10), 1886–1896. <https://doi.org/10.1039/d1lc00101a>

Yang, H., Fan, S.-K., & Hsu, W. (2008). Connecting interface for modularization of digital microfluidics. *Microfluidics, BioMEMS, and Medical Microsystems VI*, 6886(February 2008), 68860L. <https://doi.org/10.1117/12.765652>

--- Thanks for the valuable references related to this work. We have added them in the revised manuscript. Please refer to references 24, 25, 54, and 55 in the manuscript and references 3 and 4, in the SI.

REFERENCES

1. Fobel, R., Fobel, C. & Wheeler, A. R. DropBot: An open-source digital microfluidic control system with precise control of electrostatic driving force and instantaneous drop velocity measurement. *Appl. Phys. Lett.* **102**, 193513 (2013).
2. Alphonsus H. C., Ng, M., Chamberlain, D., Situ, H., Lee, V. & Wheeler, A. R. Digital microfluidic immunocytochemistry in single cells. *Nat Commun* **6**, 7513 (2015).
3. Ruan, Q., Ruan, W., Lin, X., Wang, Y., Zou, F., Zhou, L., Zhu, Z. & Yang, C. Digital-WGS: Automated, highly efficient whole-genome sequencing of single cells by digital microfluidics. *Sci. Adv.* **6**, 5597 (2020).
4. Gao, J., Liu, X., Chen, T., Mak, P. I., Du, Y., Vai, M. I., Lin, B. & Martins, R. P. An intelligent digital microfluidic system with fuzzy-enhanced feedback for multi-droplet manipulation. *Lab Chip* **13**, 443-451(2013).
5. Chen, T., Jia, Y., Dong, C., Gao, J., Mak, P. I. & Martins, R. P. Sub-7-second genotyping of single-nucleotide polymorphism by high-resolution melting curve analysis on a thermal digital microfluidic device. *Lab Chip* **16**, 743-752 (2016).
6. Li, H., Shen, R., Dong, C., Chen, T., Jia, Y., Mak, P. I. & Martins, R. P. Turning on/off satellite droplet ejection for flexible sample delivery on digital microfluidics. *Lab Chip* **20**, 3709-3719 (2020).
7. Alistar, M. & Gaudenz, U. OpenDrop: An Integrated Do-It-Yourself Platform for Personal Use of Biochips. *Bioengineering* **4**, 45 (2017).
8. Yang, H., Fan, S. K. & Hsu, W. Connecting interface for modularization of digital microfluidics. *Microfluidics, BioMEMS, and Medical Microsystems VI*, 6886 (2008). <https://doi.org/10.1117/12.765652>
9. Perry, J. M., Soffer, G., Jain, R. & Shih, S. C. C. Expanding the limits towards 'one-pot' DNA assembly and transformation on a rapid-prototype microfluidic device. *Lab Chip* **21**, 3730-3741 (2021).
10. Repetto, G., Peso, A. & Zurita, J. L. Neutral red uptake assay for the estimation of cell viability/cytotoxicity. *Nat Protoc* **3**, 1125–1131 (2008).

11. Sirenko, O., Hesley, J., Rusyn, I. & Cromwell, E. F. High-Content Assays for Hepatotoxicity Using Induced Pluripotent Stem Cell–Derived Cells. *Assay Drug Dev Techn* **12**, 43-54 (2014).
12. Drug screening of cancer cell lines and human primary tumors using droplet microfluidics. Wong, A. H., Li, H., Jia, Y., Mak, P. & Martins, R. P., et al. *Sci Rep-Uk* **7**, 9109 (2017).
13. Eduati, F., Utharala, R., Madhavan, D., Neumann, U. P., Longerich, T., Cramer, T. & Merten, C. A. A microfluidics platform for combinatorial drug screening on cancer biopsies. *Nat Commun* **9**, 2434 (2018).
14. Huang, L., Zhang, X., Feng, Y., Liang, F. & Wang, W. High content drug screening of primary cardiomyocytes based on microfluidics and real-time ultra-large-scale high-resolution imaging. *Lab Chip* **22**, 1206-1213 (2022).
15. Shim, J., Cristobal, G., Link, D., Thorsen, T., Jia, Y. W., Piatelli, K. & Fraden, S. Control and Measurement of the Phase Behavior of Aqueous Solutions Using Microfluidics. *J Am Chem Soc* **129**, 8825-8835 (2007).
16. Selimovic, S., Jia, Y. W. & Fraden, S. Measuring the Nucleation Rate of Lysozyme Using Microfluidics. , 1808-1810 (2009).
17. Werner, E. M., Lam, B. X. & Hui, E. E. Phase-Optimized Peristaltic Pumping by Integrated Microfluidic Logic. *Micromachines* **13**, 1784 (2022).
18. Srigunapalan, S., Eydelnant, I. A., Simmons, C. A. & Wheeler, A. R. A digital microfluidic platform for primary cell culture and analysis. *Lab Chip* **12**, 369-375(2012).
19. Deleersnijder, D., Callemeyn, J., Arijs, I., Naesens, M., Craenenbroeck, A. H. V., Lambrechts, D. & Sprangers, B. Current Methodological Challenges of Single-Cell and Single-Nucleus RNA-Sequencing in Glomerular Diseases. *J Am Soc Nephrol.* **32**, 1838–1852 (2021).
20. Thiesse, P., Hany, M. A., Combaret, V., Ranchère-Vince, D., Bouffet, E., Bergeron, C. Assessment of percutaneous fine needle aspiration cytology as a technique to provide diagnostic and prognostic information in neuroblastoma. *Eur J Cancer* **36**, 1544-1551(2000).
21. Jaggupilli, A., Elkord, E. Significance of CD44 and CD24 as Cancer Stem Cell Markers: An Enduring Ambiguity. *Clin Dev Immunol*, 1-11 (2012).
22. Kuchma, M. D., Kyryk, V. M., Svitina, H. M., Shablii, Y. M., Lukash, L. L., Lobyntseva, G. S. & Shablii, V. A. Comparative Analysis of the Hematopoietic Progenitor Cells from Placenta, Cord Blood, and Fetal Liver, Based on Their Immunophenotype. *Biomed Res Int*, 2015.
23. Lee, D. W., Choi, Y. S., Seo, Y. J., Lee, M. Y., Jeon, S. Y., Ku, B., Kim, S., Yi, S. H. & Nam, D. H.. High-Throughput Screening (HTS) of Anticancer Drug Efficacy on a Micropillar/Microwell Chip Platform. *Anal. Chem.* **86**, 535–542 (2014).
24. Mathur, L., Ballinger, M., Utharala, R. & Merten, C. A. Microfluidics as an enabling technology for personalized cancer therapy. *Small* **16**, 1904321 (2020).

25. Shi, L., Liu, S., Li, X., Huang, X., Luo, H., Bai, Q., Li, Z., Wang, L., Du, X., Jiang, C., Liu, S. & Li, C. Droplet microarray platforms for high-throughput drug screening. *Microchim Acta* **190**, 260 (2023).

RESPONSES TO REVIEWER #3

Thanks for the reviewer's valuable suggestions that have helped us to improve the manuscript significantly. In accordance with the reviewer's suggestions, we have run more experiments on combinational drug screening on the DMF system and characterization of the cells in biopsy samples. The quantitative images for cell viability assessments, the histological images to show the cancer features, and the sequencing showing the mutations in clinical samples have been added. We also discussed more literature for comparison, and reorganized the manuscript for a better presentation. The revised parts are highlighted in blue in the manuscript and SI. Detailed responses to each comment are shown below.

Reviewer #3 (Remarks to the Author):

Zhai et al have developed a DMF device for drug testing on cancer cells.

General comments:

1. The whole manuscript needs to be thoroughly checked for grammar and typo errors to increase the readability and to avoid minor errors. Few examples though not all are given below:

a. P19: " fellow up"

b. many more.

--- We thank for the reviewer's comment. We have thoroughly checked and had a professional English editor proofread the manuscript. Hope no typos are there in the revised manuscript.

2. In this study, mostly the effect of single best drug is studied with a single cell type. However, cancers have multiple cell types and most of them require a combinatory treatment of a number of drugs. As many reports show a combinatory drug approach, the current study is very basic to assume one drug-one cell approach of cancer treatment. Kindly refer the literature¹ for a detailed study and how the current literature provides better data.

--- Thanks for your suggestions and valuable literature.

We further carried out combinational drug screening with the MDA-MB-231 breast cancer xenograft mouse model to explore the relationship between drug screening on-chip and the screening results guided tumor treatment *in vivo*. In this experiment, Doxorubicin (Dox) and Curcumol (Cur) were used as the drug models. The biopsy samples from 15 mice were used for on-chip drug screening. Guided by the on-chip screening results, the mice were sorted into three groups, with one group treated with a single effective drug, one group with a single least effective drug, and the other group with the combinational drug. Another group of 5 mice was treated with PBS solution as a control group.

Fig. R7 shows the on-chip drug screening results of Dox alone, Cur alone and the combination of 10 μ M

Dox and various concentration of Cur. As can be seen, in most of the cases, the drug combination with additional 10 μM Dox killed more cancer cells than Cur alone.

Fig. R7. On-chip single drug and combinational drug screening results of biopsy samples from 15 individual mice. The drugs are Dox, Curcamol and Dox plus Curcamol.

Fig. R8a summarizes the cell viability at the concentrations of Dox, Cur, or Dox & Cur. As can be seen from Fig. R8a, Dox alone worked better than Cur alone in most cases with lower cell viability. The combination of two drugs always shows a better effect than a single drug, with lower cell viability. Fig. R8b shows representative on-chip drug screening fluorescent images of the biopsy samples from mouse 4, mouse 9, and mouse 13.

To test the correspondence between on-chip drug screening and in-vivo drug therapy, we treated one group of mice with 10mg/kg Dox as the positive drug group, one group of mice with 10 mg/kg Cur as the negative drug group, and one group with the combination of 10 mg/kg Dox and 10 mg/kg Cur as the combination drugs group. Mouse 4, 9, and 13 represent the positive, negative, and combinational drugs groups, respectively.

Fig. R8c shows that the tumor volumes of the positive and combinational drugs groups were smaller than those of the negative drug and control groups after one month of treatment, aligning with the on-chip drug screening results. The tumor growth trend within the treatment period differed among the mice within the same group, suggesting the existence of individual differences. In the positive drug group, the tumor volume of combinational drug Dox & Cur treated mice was smaller than those treated with single drug Dox, which was consistent with the on-chip drug screening result and suggested the higher therapeutic effect for the combinational treatment.

Body weight measurement was further used to evaluate the drug toxicity. As shown in Fig. R8d, no considerable changes in body weight were observed in the mice of the negative drug and control groups. However, in the positive drug group, the body weight of the mice reduced to some extent, suggesting the potential toxicity of the effective drug. These results collectively demonstrate the reliability of the DMF platform for on-chip combinational drug screening usage.

We have added the detailed experimental and results descriptions in the manuscript (pages 9, 10) and the SI (Fig. S6).

Fig. R8. (a) On-chip drug screening results for the biopsy samples from 15 mice with Dox (10 μ M) or Curcumol (20 μ M) or Dox (10 μ M) plus Curcumol (20 μ M) treatment and the corresponding drug administration mode and therapeutic effect in vivo. (b) Representative on-chip drug screening fluorescent imaging results for the biopsy samples from mouse 4 , 9 and 13 after Dox (10 μ M) or Curcumol (20 μ M) or Dox (10 μ M) plus Curcumol (20 μ M) treatment. (c, d) The results of mice treatment. The relationship between drug administration times and (c) tumor volume, (d) mouse body weight.

3. The primary cells need to be characterized something similar to the published literature² for better understanding of readers.

--- Thanks for your valuable suggestion and literature.

To characterize the primary cells from the biopsy liver cancer samples, we ran a flow cytometry analysis. In this experiment, the combination of CD44 and CD24 was used to identify the existence of cancer stem cell (CSC) ¹. Cells with high CD44 expression and low CD24 expression were considered CSC. A high expression is indicated by the right shift of intensity peak of stained cells compared to unstained cells. As shown in Fig. R9a and Fig. R9b, the CD44 peaks of the stained cells from two biopsy samples were right-shifted from the unstained cells, indicating a high CSC expression in both samples. The CD24 peak positions of both samples had no noticeable difference for stained and unstained cells, indicating a low expression of CD24. The high expression of CD44 and low expression of CD24 demonstrated that CSCs were obtained in the biopsy samples.

We also ran an analysis of CD34, CD45, CD14, CD19, and HLA-DR, which are associated with hematopoietic and immune cells². As shown in Fig. R9a and Fig. R9b, the peaks were similar for all the markers and showed no difference for stained and unstained cells, indicating not many immune cells existed in the biopsy sample. Based on these data, we concluded that the drug screening of the biopsy sample mainly reflected the tumor cells responses to the screened drugs.

To verify the obtained cells in the biopsy of mouse models, we ran HE staining for analysis. As shown in Fig. R9c, the similar morphology of cells in the slides, with a big nuclear to cytoplasmic ratio, suggested almost all of the cells were tumor cells.

We have added the discussion in the manuscript and SI. Please refer to pages 13, 14, lines 391-408, in the manuscript and page 18 in the SI.

Fig. R9. (a, b) Flow cytometry analysis of patients#6, #7, revealed cells with high CD44 but low CD24 expression. (c) HE staining result of the biopsy samples from two mice.

Specific comments:

1. P4 line96: it was mentioned in introduction section that “ the biopsied primary tumors has never been tested for DMF”. It appears that this is the aim of the current study. However, the biopsied samples are not tested rather some cells isolated in 2D are tested, which was not in line with recent knowledge about cancer-drug testing.

--- We meant to mention that the direct drug screening of the primary tumor cells has not been tested on DMF chip. We agree with the reviewer that the description did not precisely express what we meant to say. We have revised the sentence to acknowledge the drug screening of cells from 2D culture and clarified that we were looking for direct drug screening on primary tumor cells without *in-vitro* subculture. Please refer to page 4, lines 106-107, in the manuscript.

2. P4 Line 102: the term “xenograft mouse breast cancer” is very confusing and the same meant term is used in different forms in the article. It appears as mouse breast cancer cells, which the readers might confuse to

find in actual literature. Kindly use same and uniform term throughout the manuscript.

--- We thank for the reviewer's suggestion. We have changed the related term of "xenograft mouse breast cancer" to "MDA-MB-231 breast cancer xenograft mouse model" throughout the manuscript.

3. Figure 1: the dissociated liver tissue and other xenograft tissue provides cancer cells. However, it is fully unclear the properties of such cells and whether they are cancer stem cells and what are the surface markers and growth potential. It needs to be fully characterized to understand the scientific merits of this study.

--- Thanks for your great suggestions. We have run more experiments to characterize the cell types in biopsied samples at various cancer stages. As shown in Fig. R10, CD44, and CD24 were used as markers for cancer stem cells identification. CD34, CD45, CD14, CD19, and HLA-DR were used for hematopoietic and immune cell identification. As can be seen, cancer stem cells (CSC) have been found in the biopsied samples. Immune cells and hematopoietic cells were not prominent. The existence of the CSC indicated the tumor would grow fast if not resected.

We further measured the cell viability in these samples to demonstrate their *in-vitro* growth potential for drug screening. As shown in Fig. R11, over 80% cell viability was found in all three samples. This high viability of biopsied cells promised the following drug screening on-chip results reliable.

We have added the discussion and data in the revised manuscript accordingly on pages 13 and 14, and figure 7.

Fig. R10. (a, b) Flow cytometry analysis of patients#6, #7, revealed cells with high CD44 but low CD24 expression.

Fig. R11. (a-c) Fluorescence imaging results of the cells obtained from biopsies of patient#9, #10 and #11. Green represents living cells and red represents dead cells. Scale bars are 100 μm . (d) Cell viability results of the samples from patients#9, #10, #11.

4. P6 line 155: how the cell culture spots are pre-treated with gelatin without the other parts is not clear.

--- A 2.5 μL pipette was used to dip the gelation solution onto the cell culture electrodes and let it air dry during the chip fabrication. Actually, in our later experiments with clinical samples, we found out the gelatin coating was not a must-step for successful cell culture and drug screening. However, since some experiments were gelatin-treated, we described the actual experimental conditions in the mouse experiments. Just for your reference, it is not a mandatory condition. We have added the description in the SI, on page 6, lines 152-157, for clarification.

5. Video s3 and s4: is fully not comprehensible. The authors are requested to provide live fluorescent microscopic pictures to show the cells are moving as the cells are marked with a fluorescence dye.

--- Thanks for the reviewer's suggestion. We have added another video (video_S5) demonstrating the movement of a drop containing fluorescent cells. Cells moved with the drop without any residues on the path.

6. P11 line 246: Each biopsy sample is thought to contain 10000 cells. However, in supplementary data, it is mentioned that each area was injected 20000 cells (MDA-MB-231 cell suspensions (2×10^6 cells/ml) (100 μ l)) to start with. As these data are not clear, kindly provide the quantitative data of how many cells are started and what is the end stage of number of cells in a fully grown cancer tissue.

--- We thank for the reviewer's reminder of our mistake. We injected 2×10^6 cells (MDA-MB-231 cell suspensions, 100 μ l) subcutaneously into the right flank of individual female nude mice to establish the MDA-MB-231 breast cancer xenograft mouse model. When the tumor volume of the mice increased to 100 – 300 mm^3 , we used a biopsy needle to obtain the biopsy samples with each biopsy sample contained about 10,000 cells. We have clarified this in the revised supporting information on page 6, lines 167-168.

7. Table 1: the + sign shows cell viability of 80% as written in caption. This is arbitrary and a quantitative image should be provided to each of the data.

-- We thank for the reviewer's suggestion. Please find the following Fig. R12a for quantitative results of biopsy samples from 14 individual mice with different drugs (EP, Cis, Wzb) treatments. For example, mouse #2 had the strongest response to Cis (Cell viability was 47% (Cis 40 μ M), Figure R12b), whereas mouse #14 barely responded to Cis (Cell viability was 81% (Cis 40 μ M), Figure R12b). EP appeared ineffective (-) in mouse #9 but demonstrated high effectiveness (Cell viability was 55% (EP 40 μ M), Figure R12b) in mouse #11. Most mice showed no response to Wzb. Representative images demonstrating the toxicity of the three drugs toward the primary tumor samples from mouse #6 are presented in Fig. R12c, with red fluorescent labeling indicating an increased number of dead cells at higher Cis or EP concentrations, whereas no significant change was observed in the presence of various concentrations of Wzb. These data and corresponding descriptions have been added in the manuscript for a better understanding of the results. Please refer to pages 7, 8, lines 215 to 222, and Fig. 3 in the revised manuscript.

Fig. R12. Cell viability results for the biopsy samples from 14 individual mice with different drugs (EP, Cis, Wzb) treatment (a). (b) On-chip drug screening results for the biopsy samples from 14 mice with Dox (40 μ M) or EP (40 μ M) or Wzb (40 μ M) treatment and the corresponding drug administration mode in vivo. (c), fluorescence imaging results for the biopsy samples from mouse 6 with different drugs (Cis, EP, Wzb) treatment. Green color represents live cells, red color represents dead cells. “x”, the cell toxicity was not measured due to limited amount of samples.

8. P15 line 333: the sentence “since the tumor is resected, the tumor recurrence is used as a marker” is a controversial sentence. Usually the tumor is resected only when there is no chance of recurrence and only a small percentage of tumors show recurrence. otherwise the tumor is considered not suitable for surgery. Kindly explain.

---Thanks for your comments. In mouse experiments, only a piece of biopsy sample was taken while the majority of the tumor was still in the body for us to use the tumor growth rate as an indication to demonstrate the drug treatment efficacy. However, for clinical patients, we cannot do so for ethical reasons. The experiments were conducted double-blinded, where the patients were treated solely depending on the doctors' experiences. In hospitals, CT or MRI images are normally used for the identification of new tumors. So, we decided to use them for the recurrence of the tumor to judge whether the drug was effective or not *in-vivo*.

And yes, the tumor is usually resected when it can be totally resected with a safe margin area for a low chance of recurrence. That was also the reason why the two patients in the control group without any drug treatment after surgery were still cancer-free after 6 months. For some patients like patient #2, who had received liver transplantation due to liver cancer before this surgery, the tumor was highly suspected to be from the remaining original cancer cells in the patient. When only one tumor was found at a location that could be totally resected, resection was still the first choice for therapy. However, the chance for cancer recurrence in these cases was higher than a sole early stage tumor. Given the situation of the patient, doctors decided to treat him with a target medicine after surgery with the hope to clean up the possible remaining cancer cells somewhere else. The target medicine used for this patient happened to be the effective drug according to the on-chip drug screening, rendering him in the positive group. Given the high chance of this patient having cancer recurrence, the cancer-free in 6 months relies more on the drug, not the resection of tumor.

We have revised the manuscript on page 13 for clarification.

9. In the above, the histological pictures are required to show the tumor resection and the behavior of the original cancer after resection that it is clearly resected.

--- We thank for the reviewer's suggestion. Histological image results for patients #1 to #5 are shown in the following Fig. R13. The tumor resection was marked with a yellow circle. The CT images of patients #1, #3, #4, and #5 (Patient #2 didn't take a CT image after surgery) are shown in Fig. R14. It can be seen that the tumors have been clearly resected after surgery.

We have added the results in the SI on page 23, Fig. S10 and Fig. S11.

Fig. R13. Histological image results for patients #1 to #5. Cancer cells are highlighted with yellow area.

Fig. R14 CT images of patient#1, #3, #4, #5. The yellow circle represents patient liver.

10. Table 2: what is exactly mutated in which exons. The original data needs to be deposited with analysis for readers to understand.

--- The gene mutation for the patients was detected by Whole-exome sequencing (WES). The mutation for Patient #3 was a copy number variation for the MET gene. The exact mutations for Patient #1 and Patient #5 are shown in the below Table R1. We have added the information in the SI as Fig. S9. The related original sequence file has been also added in the SI on page 19-22.

Table R1 Mutations found in Patients #1 and #5

Patient #1	Original sequence of TP53	CTACAGTACTCCCCTGCCCTCAAC A AGATGTTTTGCCAACTGGCCAAGAC	Original amino acid	Lysine
	Mutant sequence of TP53	CTACAGTACTCCCCTGCCCTCAAC C AGATGTTTTGCCAACTGGCCAAGAC	Mutant amino acid	Glutamine
	Original sequence of ALK	GGAACATCCCCAGGCTCCAAGATGG C CCTGCAGAGCTCCTTCAC TTGTTGG	Original amino acid	Alanine
	Mutant sequence of ALK	GGAACATCCCCAGGCTCCAAGATGG T CCTGCAGAGCTCCTTCAC TTGTTGG	Mutant amino acid	Valine
	Original sequence of NF1	TACACCTGTCAGCAAATTTATGGAT C GGCTGTTGTCCCTAATGGTGTGTA	Original amino acid	Arginine
	Mutant sequence of NF1	TACACCTGTCAGCAAATTTATGGAT T GGCTGTTGTCCCTAATGGTGTGTA	Mutant amino acid	Tryptophan
Patient #5	Original sequence of TP53	TCTCTGGGGTCAACGTTTGGTCTG C AGTCCGCCGAGTATCCCGTGGTGGTG	Original amino acid	Glutamate
	Mutant sequence of TP53	TCTCTGGGGTCAACGTTTGGTCTG G CGTCCGCCGAGTATCCCGTGGTGGTG	Mutant amino acid	Alanine

11. P18 line 382: the angiogenesis affected genes should be quantified and shown here. Again the previous study should give a guideline as mentioned in general remarks.

--- Thanks for the reviewer's great suggestions. We apologize that we cannot run this experiment because the clinical samples from patients #1-5 with precision medicine observation have been used up. However, from the literature, angiogenesis is fundamental for tumor growth, invasion, and metastasis¹⁰. Undoubtedly, the agents interfering with blood vessel formation can block tumor progression. Vascular endothelial growth factor (VEGF) plays a vital role in tumor angiogenesis^{11, 12}. Up to now, all the approved therapies for hepatocellular carcinoma (HCC) are molecular-targeted therapies with anti-angiogenic effects. The primary mechanism of anti-angiogenesis is to target the VEGF and its receptors¹³. We discussed the situation of patient #3 with MET upregulating and its relationship with VEGF. In future experiments, we will monitor the angiogenesis affected genes as well in addition to the drug target genes for drug sensitivity or resistance testing.

We have added the discussion in the manuscript, page 12, lines 358-363.

12. P19 line 439: as currently the 3D culture data is foremost, a general discussion section should be devoted how this should be applicable to 3D cancer organoids. It is one of the most important limitations of the current study.

--- We thank for the reviewer's suggestion. 3D culture is the foremost technique and the final target of our system development for drug screening on primary tumor cells. However, current 3D organoid models still cannot completely recapitulate the dynamic tumor environment, including fibroblasts, endothelial cells, immune cells, and extracellular matrix³. Besides the success rate of 3D organoid models varied from 15% to 90%, leaving uncertainty about applying it to drug screening for precision medicine^{4,5}. A potential applicable 3D culture strategy on DMF chip could be using matrigel to solidify a droplet on-chip for primary tumor spheroid growth in the droplet^{6,7}. Another potential way to observe the primary tumor cell responses to the drug was to use a thin slice of tumor which already has the same microenvironment as *in-vivo* tumor^{8,9}. We will explore those techniques in future work to improve our drug screening system for 3D primary tumor drug screening.

We have added the discussion on page 16, 17, lines 485-496, for better understanding and prospection.

13. Supplementary line 80: "tumors larger or smaller are discarded". This is very ambiguous and not in line with scientific publications. Kindly mention all the data discarded.

--- We thank for the reviewer's reminder. We chose the mice with tumor volume of 100 – 300 mm³ for the experiments. The mice with tumors larger than 300 mm³ (454.9496 mm³, 365.3926 mm³, 342.4013 mm³) or smaller than 100 mm³ (40.768 mm³, 89.1015 mm³, 85.0023 mm³, 97.3814 mm³, 75.504 mm³, 49.3293 mm³) were not used for the experiments. We have added the data on page 7, lines 178-181 in the SI.

14. For all data, a corresponding fluorescent images are required for readers to understand. IN this manuscript, only quantitative data is given without any real images of cells being stained in all the scenarios.

--- We thank for the reviewer's suggestion. We have put some fluorescent images in the manuscript. Please refer to Fig. 3, Fig. 5, Fig. 6 and Fig. 7.

Literature 1: <https://www.nature.com/articles/s41467-021-25921-9>

Literature 2: <https://pubs.rsc.org/en/content/articlehtml/2023/bm/d2bm01870e>

--- Many thanks for the valuable literature. They immensely helped improve the quality of the manuscript. We have added them in the references, Ref.63 and 68.

REFERENCES

1. Jaggupilli, A., Elkord, E. Significance of CD44 and CD24 as Cancer Stem Cell Markers: An Enduring Ambiguity. *Clin Dev Immunol*, 1-11(2012).
2. Kuchma, M. D., Kyryk, V. M., Svitina, H. M., Shablii, Y. M., Lukash, L. L., Lobytseva, G. S. & Shablii, V. A. Comparative Analysis of the Hematopoietic Progenitor Cells from Placenta, Cord Blood, and Fetal Liver, Based on Their Immunophenotype. *Biomed Res Int*, 2015.
3. Qu, J., Kalyani, F. S., Liu, L., Cheng, T. & Chen, L. Tumor organoids: synergistic applications, current challenges, and future prospects in cancer therapy. *Cancer Commun* **41**, 1331–1353 (2021).
4. Maru, Y., Tanaka, N., Itami, M. & Hippo, Y. Efficient use of patient-derived organoids as a preclinical model for gynecologic tumors. *Gynecol Oncol* **154**, 189-198 (2019).
5. Lee, S. H., Hu, W., Matulay, J. T., Silva, M. V., Owczarek, T. B. & Kim, K., et al. Tumor Evolution and Drug Response in Patient-Derived Organoid Models of Bladder Cancer. *Cell* **173**, 515-528 (2018).
6. Ding, S., Hsu, C., Wang, Z., Natesh, N. R., Millen, R., Negrete, M., Giroux, N., Rivera, G. O., Dohlman, A. & Bose, S. Patient-derived micro-organospheres enable clinical precision oncology. *Cell Stem Cell* **29**, 905-917 (2022).
7. Jiang, S., Zhao, H., Zhang, W., Wang, J., Liu, Y., Cao, Y., Zheng, H., Hu, Z., Wang, S. & Ma, S. An Automated Organoid Platform with Inter-organoid Homogeneity and Inter-patient Heterogeneity. *Cell Rep Med* **1**, 100161 (2020).
8. Horowitz, L. F., Rodriguez, A. D., Dereli-Korkut, Z., Lin, R., Castro, K., Mikheev, A. M., Monnat Jr., R. J., Folch, A. & Rostomily, R. C. Multiplexed drug testing of tumor slices using a microfluidic platform. *Npj Precis Oncol* **4**, 12 (2020).
9. Chakrabarty, S., Quiros-Solano, W. F., Kuijten, M. M. P., Haspels, B., Mallya, S., Lo, C. S. Y., Othman, A. & Gaio, N. A Microfluidic Cancer-on-Chip Platform Predicts Drug Response Using Organotypic Tumor Slice Culture. *Cancer Res* **82**, 510–520 (2022).
10. Ribatti, D. et al. The role of the vascular phase in solid tumor growth: a historical review. *Neoplasia*, 1999.
11. Sampat, K. R. & O'Neil, B. Antiangiogenic therapies for advanced hepatocellular carcinoma. *Oncol* **18**, 430-438 (2013).
12. Dvorak, H. F. Vascular permeability factor/vascular endothelial growth factor: a critical cytokine in tumor angiogenesis and a potential target for diagnosis and therapy. *J Clin Oncol* **20**, 4368-4380 (2002).
13. Zhu, X. D., Tang, Z. Y. & Sun, H. C. Targeting angiogenesis for liver cancer: Past, present, and future. *Genes & Diseases* **7**. 328-335 (2020).

REVIEWER COMMENTS

Reviewer #2 (Remarks to the Author):

The authors have made significant revisions to the manuscript, resulting in notable improvements over the initial version. However, my concerns regarding the novelty and impact of the study persist. While the paper highlights the advantage of requiring minimal primary cells for each test, a key aspect in using digital microfluidics, this feature has been previously demonstrated through other types of microfluidic platforms and with higher throughput (e.g., 0.1038/s41467-018-04919-w; Merten et al. 2018). Therefore, the efficacy of drug screening with low cell numbers on the digital microfluidics platform is not surprising. Furthermore, the emphasis on electrode connectivity in their device, as acknowledged by the authors, has already been presented in existing literature. It remains unclear what substantial improvement the authors have introduced, and if any, it appears to be marginal in comparison to existing concepts, which does not justify publication in Nature Communications.

Here are additional critiques:

1. The authors adequately respond to why they use “push-buttons” instead of typical automation techniques. The authors do not adequately respond to the comments about “smart electrode sharing”.
 - a. Considering the repeated bussing strategy shown by Yang where a simple pattern can be repeated on a chip sharing the same electrical signal it is obvious to most DMF practitioners that this same principle would work with more complex designs. While it is still important to show this in literature, I do not think a high impact journal like Nature Communications is an appropriate place to show this work. I would recommend that the results detailed in figure 2 be moved to the SI or reported on in a different work.
 - b. Seeing how the authors are using typical techniques commonly used in the field, I further recommend that the label “smart digital microfluidics” be amended to just “digital microfluidics.”
2. There is some confusion in relation to the curves generated by DMF vs. well plates using

105 cells? Looking at, for example, Figure S5, it seems to show two very different responses to the same increase concentration of drugs when using “optimal conditions” for each platform (i.e., 300 cells on DMF and 105 in well plates

a. Can the authors account for this?

b. I do ultimately like the approach of validating cell numbers in droplets vs well plates. But the results are confusing.

c. It would be most convincing to show some patient samples (e.g., Figure 6) using DMF and well-plate data.

3. Line 181: The authors mention the amount of cells that can be collected with a single biopsy needle. More details on the limitations of biopsies would be beneficial (e.g how many biopsies can a single patient go through, physically or legally).

4. Line 192: The authors chose the criteria of 0.1 to 0.3 cm³ for tumor selection (Mice with larger and lower sized tumors were discarded, line 178-181 SI). No explanation seem to have been provided for this.

5. Line 211: They mention that 14 mice were used for on-chip drug screening whereas in their SI Line 181, they mention 20 mice were used for the combinational drug screening. Unclear if data is missing or if this includes the discarded mice.

6. In figure 3 a, the authors show results for the viability assay results for drug testing in mouse model. Mouse #1, 8 and 9 seem to have only tested 2 of their selected 3 drugs. Also the color choice of the legend for these 3 graphs were not consistent with the others (Cis = black, red = EP, Green = Wzb)

7. Line 436: The on-chip workflow time given is around 36h while the results of their hypoxia test showed that significant cell damage occurs after 6 hours.

Drug toxicity

8. Line 244: The authors evaluated drug toxicity based on change in body weight of the mice. I would not consider this a good metric given that many factors can weight change (stress, caloric intake, variability in baseline metabolic activity, etc.) and the mice were only monitored every other day.

9. Clinical study

10. Line 376: The authors claim the based on the results of patients #1 to #3 showing no tumor recurrence, it showed that that their screening was effective. However, patient #1 and #3 did not receive any treatment at all since this was their control. Given the large impact of individual variance, sample size is too small to make such claims.

11. In their screening for HCC specimen, they only manage to identify a one potentially effective drug for one of the patients (patient 2?). A single successful hit is not enough to indicate that their system works.

Reviewer #3 (Remarks to the Author):

Thanks a lot for responding all my queries with satisfactory experiment or citing the limitation why it cannot be performed.

RESPONSE LETTER

RESPONSES TO REVIEWER #2

(Remarks to the Author):

Comment 1:

The authors have made significant revisions to the manuscript, resulting in notable improvements over the initial version. However, my concerns regarding the novelty and impact of the study persist.

While the paper highlights the advantage of requiring minimal primary cells for each test, a key aspect in using digital microfluidics, this feature has been previously demonstrated through other types of microfluidic platforms and with higher throughput (e.g., 0.1038/s41467-018-04919-w; Merten et al. 2018). Therefore, the efficacy of drug screening with low cell numbers on the digital microfluidics platform is not surprising.

Furthermore, the emphasis on electrode connectivity in their device, as acknowledged by the authors, has already been presented in existing literature. It remains unclear what substantial improvement the authors have introduced, and if any, it appears to be marginal in comparison to existing concepts, which does not justify publication in Nature Communications.

We agree with the reviewer's opinion that the drug screening had been run on channel-based microfluidics. However, the main drawback of wasting some biopsy samples for channel-based microfluidics makes it a hot potato, considering the limited precious biopsy samples. In contrast, the DMF system can manipulate a single droplet with precise control to use up all the biopsied cells. The convenient operation of DMF without the pumps or valves, as in channel microfluidics, also validates the advantages of DMF. In this work, we used DMF to run experiments on drug screening based on the biopsy samples from MDA-MB-231 breast cancer xenograft mouse model and liver cancer specimens from patients, demonstrating tumor suppression in mice/patients treated with drugs that were screened to be effective on individual primary tumor cells.

Compared with other digital microfluidic devices, unlike the proof-of-principle demonstration of the DMF functions in various applications in literature, the digital microfluidic device presented in this paper is the first completely integrated portable equipment that functions in animal facilities and hospitals. None of the existing channel-based or electric-based digital microfluidic systems has achieved this. We have clarified this novelty in the revised manuscript on pages 3-4, lines 91-95 and page 15, lines 434-437.

Specific Critiques:

The authors adequately respond to why they use “push-buttons” instead of typical automation techniques. The authors do not adequately respond to the comments about “smart electrode sharing”.

- a. Considering the repeated bussing strategy shown by Yang where a simple pattern can be repeated on a chip sharing the same electrical signal it is obvious to most DMF practitioners that this same principle would work with more complex designs. While it is still important to show this in literature, I do not think a high impact journal like Nature Communications is an appropriate place to show this work. I would recommend that the results detailed in figure 2 be moved to the SI or reported on in a different work.

We thank the reviewer for your valuable suggestion. The novelty of this work stays in the integration of every piece, including the hardware design, DMF chip design, software control, the tumor cell preparation, drug screening on-chip, and data analysis, for a final functional portable device that can work on-site in animal facilities or hospitals for precision medicine. We agree that the electrode-sharing strategy is a trick to minimize the device and make it portable and it should not present as a key improvement. We have moved the Fig. 2 and its results to SI (page 3, 5 and 15).

- b. Seeing how the authors are using typical techniques commonly used in the field, I further recommend that the label “smart digital microfluidics” be amended to just “digital microfluidics.”

We thank the reviewer’s suggestion. We have amended the label “smart digital microfluidics” to “digital microfluidics” in the revised manuscript.

There is some confusion in relation to the curves generated by DMF vs. well plates using 10⁵ cells? Looking at, for example, Figure S5, it seems to show two very different responses to the same increase concentration of drugs when using “optimal conditions” for each platform (i.e., 300 cells on DMF and 10⁵ in well plates.

- a. Can the authors account for this?

We thank the reviewer for pointing out this. We just noticed that we used a different fitting rule for the off-chip data to get the least square fitting lines, causing the cell viability to be higher than 1 at low drug concentration, which is unreasonable. The previous difference comes from the different fitting rules for on-chip and off-chip analysis. We replotted the data by simply connecting the data points instead of fitting them into a curve for a better comparison. As shown in Fig. S6, there are subtle different responses for on-chip and off-chip to the same increased concentration of drugs, which may be attributed to the different platforms. However, the IC₅₀ for on-chip and off-chip were both around 25 μM under optimal conditions, validating the on-chip drug screening indication. We have revised Fig. S6 and the corresponding description on page 17 in SI.

b. I do ultimately like the approach of validating cell numbers in droplets vs well plates. But the results are confusing.

We are sorry for causing confusion due to our wrong data analysis method. We have corrected it in Fig. S6.

c. It would be most convincing to show some patient samples (e.g., Figure 6) using DMF and well-plate data.

Thanks for the suggestion. However, we do not have enough primary tumor cells for drug screening in well-plate. Limited amount of biopsy sample is the key reason why we developed a DMF system for drug screening of primary tumor cells on-chip, which needs 300 times fewer cells than off-chip. Next time, when we have enough cells from big tumor resections, we will run the comparison on-chip and off-chip.

Line 181: The authors mention the amount of cells that can be collected with a single biopsy needle. More details on the limitations of biopsies would be beneficial (e.g how many biopsies can a single patient go through, physically or legally).

We are not sure about law restrictions on the biopsy process. Physically, less than three biopsies are tolerable to a patient. However, more biopsy operations mean a higher risk for tumor metastasis, which is unfavorable.

Line 192: The authors chose the criteria of 0.1 to 0.3 cm³ for tumor selection (Mice with larger and lower sized tumors were discarded, line 178-181 SI). No explanation seem to have been provided for this.

We chose this range for easy biopsy operation and the follow-up precision medicine monitoring. A tumor smaller than 0.1 cm³ is too small to fully fill in the biopsy needle grove. The number of tumor cells would not be enough for three drug screenings. A tumor bigger than 0.3 cm³ may grow too big if the follow-up drug treatment is inefficient. Mice with tumor bigger than 2 cm³ must be mercy killed according to our university ethics rules. They may not last the whole observation period we designed. We have added the explanation in SI on page 8, line 216-221.

Line 211: They mention that 14 mice were used for on-chip drug screening whereas in their SI Line 181, they mention 20 mice were used for the combinational drug screening. Unclear if data is missing or if this includes the discarded mice.

For the single drug screening, we had two groups of mice, with 7 mice in each group. One group received drug Cis, and one group received Wzb; totally 14 mice were drug-treated. Another group of 7 mice was run in parallel without drug treatment as a control. So a total of 21 mice were investigated.

For the combinational drug screening, we needed to run three groups for drug treatment (one with Dox, one with Cur, and one with both). 7 mice in each group are ideal. Unfortunately, we did not get so many mice with required tumor sizes. So, we put 5 mice in each group. Another group of 5 mice was run as a control.

A total of 20 mice were investigated in this experiment.

Thanks for pointing out this. We have clarified this in the Methods section on page 10, 11 in SI.

In figure 3 a, the authors show results for the viability assay results for drug testing in mouse model. Mouse #1, 8 and 9 seem to have only tested 2 of their selected 3 drugs. Also the color choice of the legend for these 3 graphs were not consistent with the others (Cis = black, red = EP, Green = Wzb)

We thank the reviewer's comment. The amount of biopsy samples from mouse #1, 8, 9 were too little, which may be due to the half-filled biopsy needles or centrifugation process. The cells were enough for only two drug screenings. Moreover, thanks for pointing out the color issue. We have adjusted the color choice of the legend for these 3 graphs to be consistent with the others (Cis = black, red = EP, Green = Wzb) in the revised manuscript. Please refer to page 6, line 180-182.

Line 436: The on-chip workflow time given is around 36h while the results of their hypoxia test showed that significant cell damage occurs after 6 hours

The entire workflow includes tumor sample acquisition after surgery, cell dissociation, on-chip droplet operation, on-chip cell culture for drug screening, and data analysis, which in total takes around 36 hours. The cells experience hypoxia only in the first step, where the tumor has been dissected from a patient but not dissociated yet. Cell dissociation and cell culture are all in the incubator. Cell damage can be observed after 6 hours if the sample was left untreated after surgery, so the fresher the tumor samples, the better the results. This is also why we built the device as a portable one that can be taken to animal facilities or hospitals to start experiments using the freshest samples. We have clarified it on pages 13-14, lines 404-411.

Line 244: The authors evaluated drug toxicity based on change in body weight of the mice. I would not consider this a good metric given that many factors can weight change (stress, caloric intake, variability in baseline metabolic activity, etc.) and the mice were only monitored every other day.

Yes, many factors may change weight (stress, caloric intake, variability in baseline metabolic activity, etc.). However, in this research, all the mice were 6-week-old female mice. All the groups, including the control group although they were not drug treated, went through the same biopsy process and drug injection process (the control group was injected with PBS buffer) and were fed the same food in a specific-pathogen-free (SPF) facility at 23-25°C on a 12-h light/dark cycle. All these potential factors were controlled to ensure the drug was the only factor that varies among mice. As reported, change in the body weight of the mice is widely used to evaluate drug systematic toxicity.^{1,2} So, we followed the same criteria before we found another more reliable approach.

Line 376: The authors claim the based on the results of patients #1 to #3 showing no tumor recurrence, it showed that that their screening was effective. However, patient #1 and #3 did not receive any treatment at all since this was their control. Given the large impact of individual variance, sample size is too small to

make such claims.

We thank the reviewer's comments. In this work, we would like to show that the *in-vitro* drug screening of primary tumor cells on a DMF chip can predict *in-vivo* precision medicine. In mice models, we have validated it with different batches of mice, different types of drugs and drug combinations under strict controls. With clinical samples, we have successfully obtained the tumor cells and run drug screening on-chip. All 5 clinical sample drug screening results have been validated with exon sequencing. The consistency of *in-vitro* primary tumor drug screening with sequencing can already demonstrate the possibility of using primary tumor drug screening as an alternative to precision medicine prediction. We agree with the reviewer that the sample size for the drug treatment experiment was small due to the double-blinded experiment design, which would leave some space for uncertainty. We have revised the claims to accommodate the uncertainty and future work to have more samples and more types of cancers for validation. Please refer to page 14, line 424-425, and page 15, line 438-439.

In their screening for HCC specimen, they only manage to identify a one potentially effective drug for one of the patients (patient 2?). A single successful hit is not enough to indicate that their system works.

In response to the above question, we agree that the sample size of clinical treatment was small. We have revised the claims and discussed the drawbacks of this work on page 15, line 438-439. However, the other experiments on mice and sequencing of clinical samples have indicated that our system works well for *in-vitro* primary tumor cell drug screening for precision medicine prediction.

References

1. Li, F., Lu, J., Liu, J. et al. A water-soluble nucleolin aptamer-paclitaxel conjugate for tumor-specific targeting in ovarian cancer. *Nat Commun* 8, 1390 (2017).
2. Lee, Y.G., Chu, H., Lu, Y. et al. Regulation of CAR T cell-mediated cytokine release syndrome-like toxicity using low molecular weight adapters. *Nat Commun* 10, 2681 (2019).

REVIEWERS' COMMENTS

Reviewer #2 (Remarks to the Author):

The authors have addressed all the comments from the reviewers. The manuscript is ready to be published.